



# The Glacial Paleolandscapes of Southern Africa:
# the Legacy of the Late Paleozoic Ice Age

Pierre Dietrich[1,2,3*], François Guillocheau[1], Guilhem A. Douillet[2], Neil P. Griffis[4], Guillaume Baby[5], Daniel P. Le Héron[6], Laurie Barrier[7], Maximilien Mathian[8], Isabel P. Montañez[9], Cécile Robin[1], Thomas Gyomlai[1,7], Christoph Kettler[6], & Axel Hofmann[3]

[1] Univ Rennes, CNRS, Géosciences Rennes, UMR 6118, 35000 Rennes, France

[2] Institut für Geologie, Universität Bern, Baltzerstrasse 1+3, Bern, CH 3012, Switzerland

[3] Department of Geology, Auckland Park Kingsway Campus, University of Johannesburg, Johannesburg, 2006, South Africa

[4] United States Geologic Survey, Geology Geochemistry and Geophysics Science Center, Lakewood, CO 80225, USA

[5] Physical Science and Engineering Division, King Abdullah University of Science and Technology, 15  Thuwal, Saudi Arabia

[6] Department of Geology, University of Vienna, Vienna, Austria

[7] Université Paris Cité, Institut de Physique du Globe de Paris, CNRS, UMR 7154, Paris, France

[8] Université de la Nouvelle-Calédonie, ISEA, EA 7484, BPR4, 98851, Noumea, New Caledonia, France

[9] Department of Earth and Planetary Sciences, University of California, Davis, California 95616, USA
*Correspondence to: Pierre Dietrich (pierre.dietrich@univ-rennes.fr)*

**Keywords:** paleolandscapes, glacial erosion, Africa, Late Paleozoic Ice Age, paleofjords

## Abstract

The modern relief of Southern Africa is characterised by stepped plateaus bordered by escarpments. This morphology is thought to result from stepwise uplift and ensuing continental-scale erosion of the region as it rode over Africa's mantle 'superplume' following the break-up of Gondwana, i.e. since the mid-Mesozoic. We demonstrate in this contribution that this modern morphology of Southern Africa is in fact largely inherited from glacial erosion associated to the Late Paleozoic Ice Age (LPIA) that 30  occurred between 370 and 260 Myr ago, during which Gondwana – which included Southern Africa – was covered in thick ice masses. Southern Africa hosts vast (up to $10^6$ km²) and thick (up to 5 km) sedimentary basins ranging from the Carboniferous, represented by glaciogenic sediments tied to the LPIA, to the Jurassic-Cretaceous. These basins are separated by intervening regions largely underlain by Archean to Paleoproterozoic cratonic areas that correspond to paleohighlands that preserve much of



the morphology that existed when sedimentary basins formed, and particularly glacial landforms. In this contribution, we review published field and remote data and provide new large-scale interpretation of the geomorphology of these paleohighlands of Southern Africa. Our foremost finding is that over Southern Africa, vast surfaces, tens to hundreds of thousands km² (71.000-360.000 km²) are exhumed glacial landscapes tied to the LPIA. These glacial landscapes manifest in the form of cm-scale striated

pavements, m-scale fields of *roches moutonnées*, whalebacks and crag-and-tails, narrow gorges cut into high-standing mountain ranges, and km-scale planation surfaces and large U-shaped valleys, overdeepenings, fjords and troughs up to 200 km in length. Many modern savannahs and desertic landscapes of Southern Africa are therefore relict glacial landscapes and relief ca. 300 Myr old. These exhumed glacial relief moreover exerts a strong control on the modern-day aspect of the

geomorphology of Southern Africa as (1) some escarpments that delineate high-standing plateaux from valleys and coastal plains are inherited glacial relief in which glacial valleys are carved, (2) some hill or mountain ranges already existed by LPIA times and were likely modelled by glacial erosion, and (3) the drainage network of many of the main rivers of Southern Africa is funnelled through ancient glacial valleys. This remarkable preservation allowed us to reconstruct the paleogeography of

Southern Africa in the aftermath of the LPIA, consisting of highlands over which ice masses nucleated and from which they flowed through the escarpments and toward lowlands that now correspond to sedimentary basins.

Our findings therefore indicate that glacial landforms and relief of continental-scale can survive over tens to hundreds of million years. This preservation and modern exposure of the glacial

paleolandscapes were achieved through burial under piles of Karoo sediments and lavas over ca. 120 to 170 million years and a subsequent exhumation since the middle Mesozoic owing to the uplift of Southern Africa. Owing to strong erodibility contrasts between resistant Precambrian bedrock and softer sedimentary infill, the glacial landscapes have been exhumed and rejuvenated.

We therefore emphasise the need of considering the legacy of glacial erosion processes and the

resulting presence of glacial landscapes when assessing the post-Gondwana-breakup evolution of Southern African topography and its resulting modern-day aspect, as well as inferences about climate changes and tectonic processes. Finally, we explore the potential pre-LPIA origin for some of the landscapes. In the Kaoko region of northern Namibia, the escarpments into which glacial valleys are carved may correspond to a reminiscence of the Kaoko Pan-African Belt, whose crustal structures

were either reactivated or where relief persisted since then. In South Africa, the escarpment bordering the paleohighland corresponds to crustal-scale faults that might have been reactivated during LPIA by subsidence processes. These inherited morphological or crustal features may have been re-exploited and enhanced by glacial erosion during the LPIA, as it is the case for some Quaternary glacial morphology.



## 1. **Introduction**

Glacial erosion processes profoundly shape the relief of glaciated continents and continental shelfs. For instance, planation surfaces, U-shaped valleys and fjords, overdeepenings and cross-shelf troughs that dominate the current morphology of northern North America, Greenland, Scandinavia and Antarctica largely result from glacial erosion occasioned by the expansion and demise of Cenozoic
and Quaternary ice sheets (see contributions in this special issue; Sugden and Denton, 2004; Jamieson et al., 2008; Steer et al., 2012; Medvedev et al., 2013; Herman et al., 2015; Dowdeswell et al., 2016; Egholm et al., 2017; Paxman et al., 2018, 2019; Bernard et al., 2020; Couette et al., 2022; Vérité et al., 2021, 2023, 2024). Southern Africa was also covered in continental-scale ice masses, twice over the Phanerozoic during icehouse climate periods, on the occasion of the Ordovician (445 Myr ago) and
Late Paleozoic ice ages (370-260 Myr ago, Ghienne et al., 2007; Le Heron et al., 2009; Montañez, 2021). Considering the antiquity of these ice ages, their contribution to the modern-day morphology of southern Africa is generally neglected. Indeed, the morphological footprint of glacial erosion processes are generally considered to be largely transient at geological time scales, prone to be rapidly erased over a few million years (Prasicek et al., 2015). The long-term evolution of the Southern
African topography and the resultant modern-day landscapes are therefore viewed as originating from erosion-sedimentation processes and lithospheric uplifts in response to tectonic and non-glacial climate forcings over the Cenozoic and the Mesozoic (Burke and Gunnell, 2008; Feakins and Demenocal, 2012; Kamp and Owen, 2013; Paul, 2021). The high-standing plateaux, pediments and coastal plains separated by intervening escarpments and valleys that characterize the peculiar
morphology of southern African are indeed interpreted as mostly originating from Atlantic rifting and continental break-up processes (Dauteuil et al., 2013; Salomon et al., 2015). Such phenomenon are denudation, fluvial erosion and scarp retreat paced by anorogenic uplifts tied to the polyphase activity of the African mantle plume since 130 Myr (Moucha and Forte, 2011; Braun et al., 2014; Goudie and Viles, 2015; Mvondo Owono et al., 2016; Braun, 2018; Guillocheau et al., 2018; Margirier et al.,
2019). Yet, many regions of southern Africa bear planation surfaces, U-shaped valleys and m- to km-scale landforms that happen to be glacially-scoured paleorelief tied to the Late Paleozoic Ice Age (Lister, 1987; Visser, 1987a; Andrews et al., 2019; Dietrich and Hofmann, 2019; Le Heron et al., 2019, 2022, 2024; Dietrich et al., 2021), suggesting that the contribution of glacial erosion processes in shaping the modern morphology of Southern Africa has largely been underestimated.


Here we test the idea that the current relief and landscapes of Southern Africa are largely inherited from late Paleozoic glacial erosion processes. For doing so, we present new field and remote sensing geomorphic observations along with a compilation of existing studies and geological and GIS-based mapping, which we integrate with sedimentologic studies to test the origin of landscapes scattered across southern Africa and apprise their origin (Fig. 1). This combined approach revealed the
presence of vast ($10^3$-$10^5$ km²) glacial paleolandscapes carved during the Late Paleozoic Ice Age or



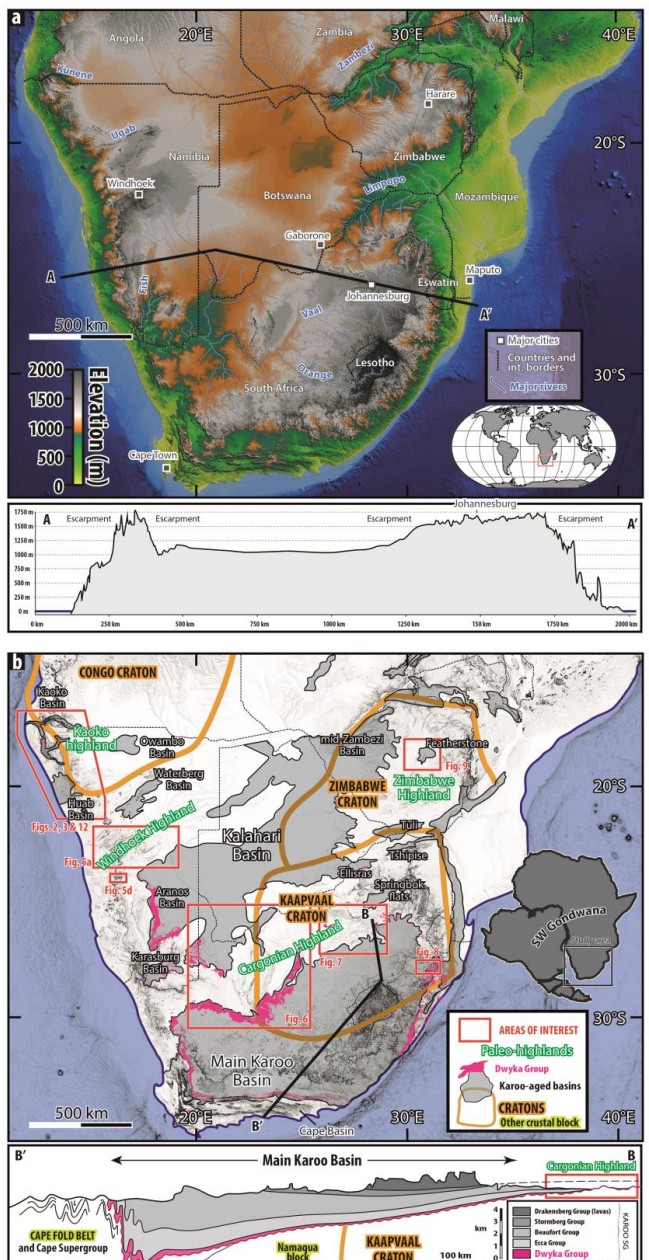

Fig. 1: (a) Modern relief of Southern Africa shown by Digital Elevation Model (DEM) from Shuttle Radar Topographic Mission (https://www2.jpl.nasa.gov/srtm/) along with major river networks, international borders and main cities. The transect highlights the high-standing plateaus. (b) Southern Africa with regions of interest discussed in the text shown by red frame. The Archean to Paleoproterozoic Congo, Kaapvaal and Zimbabwe cratons are evidenced by thick orange lines and Karoo-aged sedimentary basins are represented by grey shaded area. The glaciogenic Dwyka group is represented by pink colour. Inset map shows western Gondwana formed by Africa and South America and the four paleohighlands discussed in the text are evidenced in green. Transect displays the thickness and sedimentary succession of Main Karoo Basin (MKB) of South Africa, the glaciogenic Dwyka Group in pink, the glacial erosion surface (wavy pink line) at the base of the Karoo Supergroup and the underlying basement structure (cratons vs. accreted terranes). Transect modified after Johnson et al., 1996 and Karoo-aged basins after Catuneanu et al., 1998.



terrains forming the three cratons situated in Southern Africa, the Kaapvaal, Zimbabwe and Congo cratons. Our study therefore revives the concept of ancestral landsurfaces over Southern African cratons (the 'Gondwana Surface' of King, 1948, 1949, 1982; see also Twidale, 2003; Doucouré and de Wit, 2003; Guillocheau et al., 2018) and even extend it further back in time. Secondly, compiling thermochronometrical and stratigraphic data, we address the preservation of these relict glacial landscapes -how they escaped being erased for over hundreds of millions of years- through burial and their renewal through continental-scale erosion, owing to strong erodibility contrast between the substrate and sedimentary infill of these paleorelief. We also emphasise the need to consider ancient glacial erosion morphological features as a major component of the modern-day African landscapes to reconstruct long-term topographic evolution of the African surfaces and to quantify source-to-sink budgets, themselves serving as inferring past climate changes, base level variations and tectonic processes (Gilchrist et al., 1994; Rouby et al., 2009a; Kamp and Owen, 2013; Mvondo Owono et al., 2016; Baby et al., 2018a, 2018b, 2020a; Grimaud et al., 2018).

## 2. State-of-the-Art: the relief and geology of Southern Africa and the record of the ice ages

### 2.1. General physiography

We refer here to Southern Africa as a *ca.* 4.000.000 km² region shared by South Africa, Namibia, Botswana, Zimbabwe, Mozambique, Lesotho and Eswatini (Fig. 1a). The morphology of this region is characterized by high-standing plateaus lying above 1000 m.a.s.l. and frequently above 2000 m.a.s.l. (Fig. 1a). These plateaus are surrounded by steep escarpments leading downward to stepped plateaus (planation surfaces) of lower elevation and ultimately to the coastal plains (see Braun et al., 2014; Guillocheau et al., 2018; Baby et al., 2018a, 2018b, 2020 and references therein for details). The escarpments are dissected by valleys focusing the drainage network, such as the Orange-Vaal-Fish and Ugab and Kunene rivers to the West in the Atlantic, Zambezi and Limpopo to the East in the Indian Ocean and endorheic system at the center (Baby et al., 2020).

### 2.2. Geological setting

Southern Africa is rooted by three Archean to Paleoproterozoic cratons -the Kaapvaal, Zimbabwe and Congo cratons (Fig. 1b)- that amalgamated via younger terranes during orogenic events throughout the Proterozoic (Tankard et al., 2009; Begg et al., 2015; Torsvik and Cocks, 2016). As part of SW Gondwana during the Paleozoic and early Mesozoic, Southern Africa was mostly subsiding which permitted the deposition of thick sediment piles that form the Cape and Karoo Supergroups. Later,



during the late Mesozoic and Cenozoic, after the dislocation of Gondwana, anorogenic uplift related to Indian and Atlantic oceans break-ups and post-rift mantle dynamics led to the inversion of the sedimentary basins and widespread, continental-scale erosion and planation (Veevers et al., 1994; Lithgow-Bertelloni and Silver, 1998; Moulin et al., 2010; Braun et al., 2014; Linol and Wit, 2016a; Braun, 2018; Guillocheau et al., 2018). With the notable exception of the Cape Fold Belt, situated at the southern tip of South Africa, an orogen tied to the subduction of the Panthalassic Ocean that initiated during the Permian-Triassic (Hansma et al., 2016), no orogenic event affected the rest of Southern Africa throughout the Phanerozoic.

Over the basement lie vast, $10^3$-$10^6$ km², sedimentary basins referred to as 'Karoo-aged basins'. They are named after the Main Karoo Basin (MKB) of South Africa, the thickest, largest (ca. 700.000 km²), and most studied of these basins, investigated since at least the mid-XIX$^{th}$ (see Linol and de Wit, 2016). Together, the Karoo-aged basins of southern Africa cover an area of ca. 1.600.000 km², among which ca. 800.000 km² are subcrop, mostly covered by younger sediments of the Kalahari Desert (Fig. 1b, Catuneanu et al., 2005; Haddon, 2005). The volcano-sedimentary pile that forms these Karoo-aged basins -the Karoo Supergroup- can be up to 5 km thick and ranges in age from Carboniferous to Jurassic (in South Africa and Zimbabwe) or Cretaceous (in northern Namibia) (Stratten, 1977; Smith, 1990; Smith et al., 1993a; Johnson et al., 1996; Catuneanu et al., 2005; Milani and De Wit, 2008; Franchi et al., 2021). Depositional environments within the Karoo Supergroup range, from base to top, from glacial (the Dwyka Group), marine (Ecca & Beaufort groups), continental and aeolian (Stormberg Group) and finally subaerial lava outpouring (Drakensberg and Etendeka Groups) (Fig. 1b; Johnson et al., 1996). As such, these basins record the intracratonic nature of Southern Africa, the Cape orogeny and the Drakensberg and Etendeka lavas topping the Karoo Supergroup related to the break-up of Gondwana. The Main Karoo and Kalahari basins have been interpreted as a foreland to the Cape orogeny (Visser, 1992, 1993; Johnson et al., 1997; Catuneanu et al., 1998; Catuneanu, 2004; Isbell et al., 2008), or as resulting from a deflection of the lithosphere due to mantle flow coupled to the adjacent subduction of the Panthalassic ocean (Pysklywec and Mitrovica, 1999). Stratigraphically below the MKB lies the Cape Basin formed by the Cape Supergroup, ca. 3.000 m-thick and ranging in age from the Cambrian to the lower Carboniferous, deposited over a continental platform, today cropping out at the southern tip of South Africa over ca. 90.000 km², and highly deformed during the Cape orogeny (Fig. 1b, Streel and Theron, 1999; Shone and Booth, 2005; Thamm and Johnson, 2006; Tankard et al., 2009).

### 2.3. The record of the ice ages

During the Paleozoic, Southern Africa as part of SW Gondwana experienced two distinct and extensive glaciations, with ice extent of $10^5$-$10^7$ km²: the short-lived Late Ordovician episode (445 Ma,



Deynoux and Ghienne, 2004; Ghienne et al., 2007; le Heron and Dowdeswell, 2009; Le Heron et al., 2009) and the protracted Late Palaeozoic Ice Age, hereafter referred as LPIA (ca. 370-260 Ma, Isbell et al., 2012, 2021; Montañez and Poulsen, 2013; Griffis et al., 2019a, 2019b, 2021, 2023; Montañez,

2021). Southern Africa witnesses these two glacial episodes under the form of glaciogenic sedimentary successions within the Cape and Karoo supergroups and/or relict glacial erosion features carved on the bedrock.

### 2.3.1.  The Late Ordovician glacial episode

The Ordovician glacial episode is recorded within the Cape Supergroup under the form of a thin (<50

m) layer of diamictite, corresponding to an unsorted mixture of fine- and coarse-grained sediments, named the Pakhuis Pass Formation, and interpreted as having been deposited under a flowing ice sheet (Thamm and Johnson, 2006; Blignault and Theron, 2010). A single outcrop displays a striated glacial pavement at the Pakhuis Pass of the Cederberg region (Deynoux and Ghienne, 2004). High-amplitude (50 m) folds are present below the Pakhuis Pass formation and are linked to the activity of a glacier

flowing over waterlain soft sediments (Backeberg and Rowe, 2009; Blignault and Theron, 2010, 2017; Rowe and Backeberg, 2011). No major relict glacial erosion landforms are associated to this glacial episode. Although renown for topping the iconic Table Mountain overhanging the city of Cape Town, the sedimentary record of the Ordovician glacial episode is spatially restricted to the Cape Fold Belt in South Africa (Thamm and Johnson, 2006; Ghienne et al., 2007; Fourie et al., 2010; Meadows and

Compton, 2015; Davies et al., 2020).

### 2.3.2.  The Late Paleozoic Ice Age (LPIA)

The lowermost sedimentary unit of the Karoo Supergroup, directly lying on the bedrock, is Carboniferous-Permian in age and has a glaciogenic origin, deposited by ice sheets during the LPIA (the pink layer within Fig. 1b; Visser, 1990; Johnson et al., 1996; Cairncross, 2001; Catuneanu et al.,

2005; Griffis et al., 2018, 2019a, 2021). This glaciogenic sedimentary unit is extensively present in southern Africa and crops out or has been identified through drilling in all Karoo-aged basins (Fig. 1b, Smith, 1994; Catuneanu et al., 2005). It is named Dwyka Group within the MKB after the Dwyka River crossing the Cape Mountain in the Western Cape Province (Dunn, 1886; Pfaffl and Dullo, 2023) and its equivalents within other Karoo-aged basins have different, locally-sourced names such as

Dukwi, Waterkloof, Gibeon, Malogong or Tshidzi (Haughton, 1963; Smith, 1994; Johnson et al., 1997; Modie, 2002, 2008; Catuneanu et al., 2005; Bordy, 2018). For sake of clarity, we will refer to it as the Dwyka Group throughout the manuscript, independently of the basin considered. The Dwyka Group within the MKB and the Aranos Basin of Namibia is typically several hundreds of meters thick but has been found as thin as a few cm in some other Karoo-aged basins (Visser, 1987a, 1987b, 1997;

Isbell et al., 2008; Stollhofen et al., 2008; Miller, 2011; Dietrich and Hofmann, 2019). The lithologies and facies encountered within the Dwyka Group are very diverse, but commonly consist of diamictites, clast-bearing mudstones and conglomerates, interpreted as representing various glacial-




related depositional environments, and have been the focus of a plethora of studies (Martin and
Schalk, 1959; Crowell and Frakes, 1972; Stratten, 1977; Visser, 1982, 1983, 1987a, 1987b, 1994,
1997; Visser and Kingsley, 1982; Visser and Hall, 1985; Visser and Loock, 1987; Smith et al., 1993b;
Brunn et al., 1994; Veevers et al., 1994; Johnson et al., 1996, 2006a; Von Brunn, 1996; Haldorsen et
al., 2001; Werner and Lorenz, 2006; Fielding et al., 2008; Isbell et al., 2008; López-Gamundí and
Buatois, 2010; Miller, 2011; Linol and Wit, 2016; Dietrich and Hofmann, 2019; Dietrich et al., 2019,
2021; Menozzo da Rosa et al., 2023; Fedorchuk et al., 2023; Fernandes et al., 2023). Glacial striae,
grooves and lineations carved by flowing glaciers within soft sediments are common features of the
Dwyka Group (Dietrich and Hofmann, 2019; Le Heron et al., 2019). Interestingly, these coarse-
grained deposits associated to glacial pavements have long been attributed a glacial origin (Sutherland,
1868, 1870; Dunn, 1886, 1898; Molengraaf, 1898; Cloos, 1915; Wagner, 1915; Du Toit, 1921; Pfaffl
and Dullo, 2023). Also, the glaciogenic Dwyka Group and its South American equivalent, on the other
side of the Atlantic Ocean, largely led South African geologist Alexander L. Du Toit (1878-1948) and
German geologist Henno Martin (1910-1998), at some stage director of the Geological Survey of
Namibia, to be early supporters of Wegener's theory of continental drift (Du Toit, 1921, 1927, 1933,
1937; Martin and Schalk, 1959; Martin, 1961, 1973b, 1973a; see also Haughton, 1949; Milani and De
Wit, 2008; Miller, 2011; Linol and de Wit, 2016; Pfaffl and Dullo, 2023)

The Karoo-aged basins are separated by cratonic regions over which no or very thin Dwyka
and Karoo deposits lie. Wedging out and onlap of Karoo strata against these regions and depositional
environments of the Dwyka Group pointing toward a continental ice sheet together indicate that these
'no or thin Dwyka' areas  correspond to regions of no or little sediment deposition, i.e. highlands that
already existed during the deposition of the Dwyka Group, and hereafter referred to as *paleohighlands*
(Fig. 1b, Visser, 1985, 1987a, 1987b, 1997; Smith et al., 1993a; Catuneanu et al., 1998, 2005; Isbell
and Cole, 2008). Four paleohighlands exist over Southern Africa: the Kaoko, Windhoek, Cargonian
and Zimbabwe, underlain by resistant Archean to Paleoproterozoic basement rocks that form craton
and/or shield areas (Fig. 1b). Over these paleohighland substrates are carved glacial landforms and
erosion surfaces. These surfaces, either sculped by direct glacial abrasion or eroded by meltwater,
encompass a range of landform types and shapes ranging from small ($10^{-2-0}$ m: striae and grooves) to
intermediate ($10^{0-3}$ m: *roches moutonnées*, cirques, whalebacks, and crag-and-tails), and large scale
($10^{4-6}$ m: fjords, troughs, and glacial overdeepenings) glacial forms and landsurfaces (Benn and Evans,
2010). These glacial erosion forms are the main focus of the present contribution for which a review
and new constrains are provided below (section 3 and figures 2 to 8). These glacial erosion surfaces
and paleolandsurfaces have been named 'ancestral glacial pre-Karoo peneplain' (Wellington, 1937), or
'pre-Dwyka topography' (Du Toit, 1954; von Gottberg, 1970b). It is indeed generally assumed that
they were carved during the LPIA, immediately before the deposition of the Karoo Supergroup that
started with the glaciogenic Dwyka Group, although their potential pre-LPIA origin is discussed below



(section 4.2). In the center of the Karoo-aged basins, the glacial landforms are believed to be still
covered by thick sedimentary piles of the Karoo Supergroup and therefore await renewal through
exhumation (see transect on Fig. 1b).

As no major horizontal tectonic event affected the region after the deposition of the Karoo
Supergroup (Torsvik and Cocks, 2016), except for the localized Permian-Triassic Cape orogeny,
glacial paleoreliefs have preserved their original shape and orientation and Dwyka strata lie mostly
horizontally. Slight and local tilt of the Karoo strata are due to vertical motions of the lithosphere
induced by the activity of the African superplume or isostatic processes linked to the Atlantic passive
margin.

## 2.4.    Post Gondwana-breakup history of the Southern Africa Plateau

Following the break-up of Gondwana, which milestones are the opening of the Indian and Atlantic
oceans in the early Jurassic and early Cretaceous (Frizon De Lamotte et al., 2015; Thompson et al.,
2019; Roche et al., 2021; Roche and Ringenbach, 2022), lithospheric movements were mostly vertical
in response to the African mantle 'superplume'. Pulses in the activity of this plume led to episodes of
swelling and uplift of Southern Africa. This stepwise uplift induced multiple phases (polycyclic) of
erosion and planation of the interior of the continent, including the inversion of the Cape and Karoo
basins, retreat of the passive margin escarpments and export of sediments thus produced to the
continental margins (e.g., Partridge and Maud, 1987, 2000; van der Beek et al., 2002, Braun, 2010,
2018; Braun et al., 2014; Baby et al., 2018a, 2018b, 2020; Guillocheau et al., 2018; Stanley et al.,
2021). By quantifying and budgeting onshore erosion (through geomorphology and
thermochronometry) and offshore sediment accumulation (through seismic stratigraphy), sediment
fluxes have been reconstructed which together with other inferences (assessment of sediment routing,
characterization of kimberlite pipes etc.) allowed to reconstruct the post Gondwana-breakup history of
the Southern African Plateau, as summarized thereafter.

- During the lower Cretaceous, erosion of the Southern African plateau was spatially restricted
and erosion products were funneled eastward through a proto Orange River system in the South and
eastward through the proto Zambezi-Limpopo drainage in the North (De Wit, 1999; Moore and
Larkin, 2001; Baby, 2017; Ponte, 2018; Baby et al., 2020; Stanley et al., 2021).
- A first period of accelerated denudation of the margins of the plateau followed, at ~150-120
Ma, tied to continental breakup and post-rift erosion of the rift shoulders, as indicated by
thermochronometric data (e.g., Gallagher and Brown, 1999; Brown et al., 2002; Tinker et al., 2008b;
Kounov et al., 2009, 2013; Stanley et al., 2013, 2015, 2021; Wildman et al., 2016, 2015a; Green et al.,
2017). A second pulse of denudation took place at ~100-70 Myr and coincides with acceleration of
offshore sediment flux (Walford et al., 2005; Tinker et al., 2008a; Rouby et al., 2009; Guillocheau et



al., 2012; Said et al., 2015; Baby et al., 2020). This second pulse of denudation likely resulted from the tilting of the plateau to the west that steepened the slopes across the sub-continent and enhanced a widespread and fast erosion response, notably of the passive margins escarpments (Braun et al., 2014; Baby et al., 2018b; Ding et al., 2019; Stanley et al., 2021). In this context, Wellington (1955) and Braun et al. (2014) highlighted the importance of the erodibility contrast between the soft Karoo sedimentary cover and the underlying harder basement.

- By the end of the Cretaceous, the western side of the plateau was uplifted, as suggested by offshore stratigraphic observations (Aizawa et al., 2000; Paton et al., 2008; Baby et al., 2018b) and onshore kimberlites pipes ages distribution (Jelsma et al., 2004; Braun et al., 2014). This could have resulted in a symmetrical configuration of the Southern African plateau (similar to the modern one), which would have strongly reduced the erosion potential by reducing the slope of a large portion of the sub-continent interior (Braun et al., 2014; Baby et al., 2020; Stanley et al., 2021). This scenario is supported by a drop in offshore sediment flux (Baby et al., 2020), the preservation of crater-lake sediments in ~75-65 Myr kimberlite pipes (Moore and Verwoerd, 1985; Scholtz, 1985; Smith, 1986), and the onset of the Kalahari Basin aggradation (Haddon and McCarthy, 2005).

- Thermochronometric data and offshore sediment fluxes point to limited erosion of the plateau during the Cenozoic (Stanley et al., 2021 and references therein). Recent low-temperature thermochronological data show that Cenozoic erosion focused along the present-day river valleys rather than being broadly distributed as during the Middle Cretaceous (Stanley and Flowers, 2023).

Although each pulse of uplift and therefore erosion is thought to be recorded as a planation surface, it has been suggested that very ancient, possibly Jurassic or older, landsurfaces are preserved across southern Africa, older than the initiation of uplift of southern Africa and predating the dislocation of Gondwana, and called the 'Gondwana surface' (Du Toit, 1933; King, 1949a, 1949b, 1982; Doucouré and de Wit, 2003).

## 3. Glacial paleoreliefs of Southern Africa

In the following, we describe and review the geomorphology and sedimentary infill of glacial landforms preserved across the paleohighlands of Namibia (the Kaoko and Windhoek highlands, section 3.1 and 3.2), South Africa and Botswana (the Cargonian Highland, section 3.3) and Zimbabwe (the Zimbabwe Highland, section 3.4) (Fig. 1). This is done by combining novel field and aerial/satellite observations, digital elevation model (DEM) from Shuttle Radar Topographic Mission (SRTM, https://www2.jpl.nasa.gov/srtm/) and geological maps analysis as well as an assessment of existing literature. We provide morphostratigraphic transects evidencing the presence of glacially-carved reliefs and the presence of glaciogenic Dwyka sediments based on geological maps and digital elevation models (fig. 3 to 9).



Based on these data, we provide description and interpretation of the glacial paleoreliefs and the glaciogenic sedimentary rocks occupying the glacial reliefs wherever present. For each study sites, we showcase indisputable evidences for a glacial origin (plain pink line on transects on fig. 2 to 9) or,

for suspected glacial landscapes, we provide supporting data and discuss their potential glacial origin (dotted pink line on transects on fig. 2 to 9). For these latter cases, additional field-based examinations



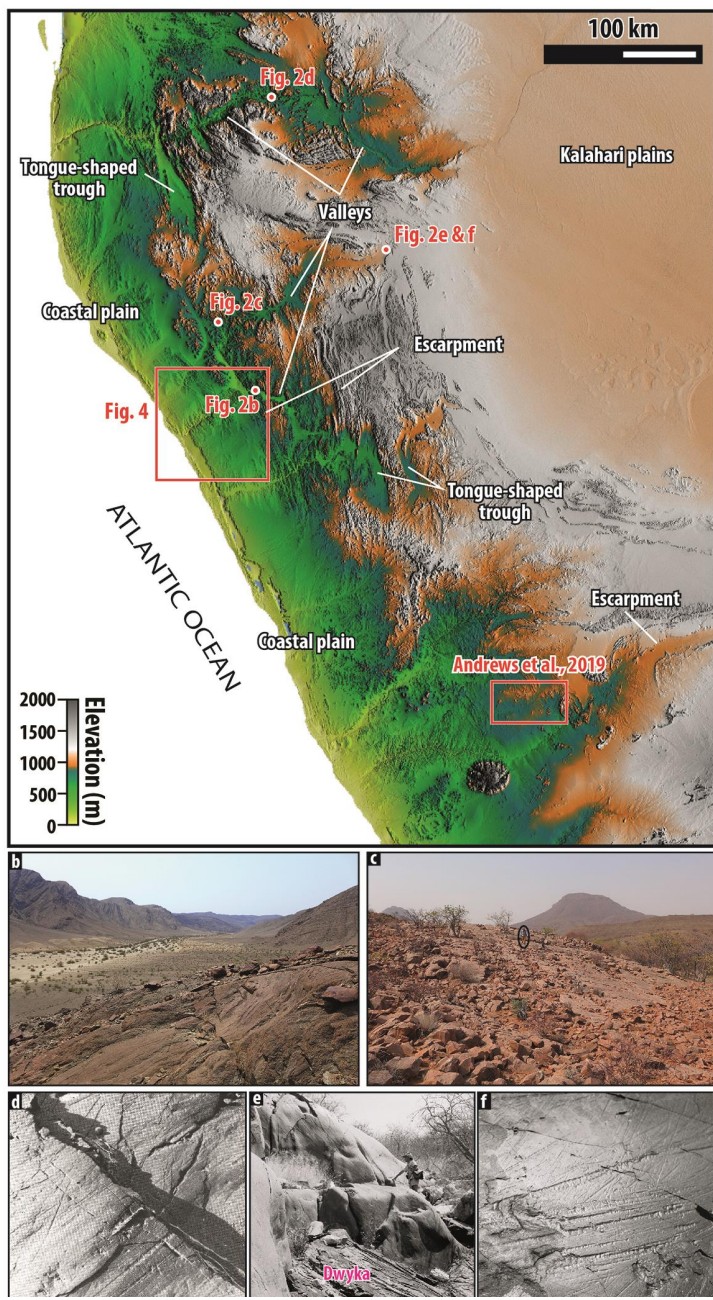

Fig. 2: (a) DEM of the Kaoko region of Northern Namibia, corresponding to the Kaoko paleohighland. The escarpments, valleys and tongue-shaped troughs discussed in the text are arrowed. Location of the pictures are also indicated. Figure 1b for location. (b) the Gomatum valley corresponds to a fjord carved during the LPIA, later sealed and exhumed in recent times. See Dietrich et al., 2021 for further details. Valley is ca. 2.5 km wide and 550 m deep. (c) A field of *roches moutonnées* and whalebacks characterized by glacial striae and grooves and polished floors covered in places by boulder pavement, evidencing a westward ice movement. Circled geologist for scale, see Le Heron et al. (2024) for details. (d) Striated floor in the Kunene valley, plucking at the joint shows ice movement from east to west. Picture from Martin, 1961; (e) Glacially polished walls and (f) floor in NE Kaoko. Pictures taken by K.E.L. Schalk, geologist Henno Martin for scale, see Miller (2011).



targeting striated pavements or other evidence for glacial activity would be required to confirm their
glaciogenic nature. Erosion and resulting landforms of mountain glaciers that existed locally during
the Quaternary will not be considered in this study (Hall and Meiklejohn, 2011; Knight and Grab,
2015).

## 3.1. The Kaoko Highland

The Kaoko region of NW Namibia is formed by a plateau that stands at ~1000 m above the sea-level
and is located at the boundary between the Congo Craton and Panafrican orogenic belts (Figs. 1 and
2). This plateau is bordered to the west by a steep escarpment leading to the Atlantic Ocean through a
gently inclined, ~50 km-wide coastal plain. Eastward, the plateau leads toward the even relief of the
Kalahari plains. On the northern half of the Kaoko region, a network of E-W-oriented valleys in which
modern rivers flow toward the Atlantic deeply dissect both the plateau and the escarpment. Here, the
escarpment is two-stepped. Tongue-shaped troughs, N-S-oriented, are also incised within the highland
or at the feet of the escarpments. These troughs either connect with the river network or are endorheic.
The southern half of the Kaoko region is characterized by a plateau separated on its western side from
the coastal plain by a single escarpment.

In the northern region of the Kaoko Highland, a network of E-W oriented valleys between the
Kunene River to the North and the Hoanib River to the South has been interpreted by Dietrich et al.
(2021) as an exhumed glacial landscape (Figure 3). These valleys are U-shaped and their floors and
subvertical flanks display abundant small-scale hard-bed glacial erosion features such as striae,
grooves, whalebacks and *roches moutonnées* (Fig. 2, see also Fedorchuk et al., 2023). Paleovalleys in
Angola that cut through the escarpment may also correspond to glacial valleys (Moragas et al., 2023).
In places, these glacial erosion features are covered with remnants of glaciogenic sediments of the
Dwyka Group (Fig. 3), including frontal and lateral moraines and glaciomarine sediments such as ice-
rafted debris scattered in shales (Fig. 2; see also Martin and Schalk, 1959; Dietrich et al., 2021;
Menozzo da Rosa et al., 2023), and fig. 16.1 in Miller (2011). These glaciogenic sediments are
typically found abutting against the valley walls (Fig. 3; Le Heron et al., 2024). Based on the relict
glaciogenic forms and associated rocks, these modern valleys were interpreted as exhumed paleofjords
whose modern U-shaped profiles reflects their original glacial morphologies (Dietrich et al., 2021, see
also Martin, 1953, 1961, 1968, 1973b, 1981; Martin and Schalk, 1959). Moreover, Dietrich et al.
(2021) demonstrated that the Purros escarpment separating the high standing plateau to the coastal
plain already existed by the LPIA as indicated by glacial striae found on the scarp (Fig. 2). The glacial
origin for the network of glacial valleys and the escarpment indicate that the Kaoko plateau already
existed by LPIA times which formed the Kaoko paleohighland. In this same region, at the downstream
end of some of the aforementioned U-shaped valleys are deep and encased bedrock canyons, such as
the Purros and Khowarib canyons. Geological maps indicate scattered glaciogenic sedimentary rocks



within these canyons (Fig. 4). We therefore suggest that these canyons forming the downstream
continuation of exhumed glacial valleys may correspond to gorges similar to those characterizing the
base of Quaternary glacial valleys found in almost all terrains that experienced repeated Quaternary
glaciations (e.g., Lajeunesse, 2014; Livingstone et al., 2017). Such an interpretation however awaits
confirmation by further sedimentological and geomorphological characterization. Moreover, the N-S-
oriented tongue-shaped troughs, such as the Omarumba-Omutirapo, the Sesfontein and Warmquelle at
the head of the Hoanib valley, and the Otjinjange (Fig. 2 & 3) were interpreted as glacial cirques by
Martin (1953, 1961, 1968, see also Hoffman et al., 2021 pages 105-106). Here though, no glaciogenic
sediments or morphological features have so far been described or reported, hindering a definitive
interpretation. To summarize, the network of valleys dissecting the northern half of the Kaoko
highland as well as the Purros escarpment are true relics of a glacial landscape; in this same region, it
remains unclear whether the interfluves between the exhumed glacial valleys, the tongue-shaped
troughs and the canyons are glacial in origin, as they might have been lowered down by erosion in
post-LPIA times

Further south, in the Huab-Uniab regions (Figs. 2 and 3), outliers of Karoo sediments and
Cretaceous Etendeka lavas topping the Karoo succession form the western part of the plateau, and rest
on highly uneven basement rocks dipping southward (transects 4 and 5 on figure 3). This volcano-
sedimentary pile, reaching 1000 m in maximum thickness to the south, thins toward the north: the
basal sedimentary units are present only in the deepest part of these basins, formed by the Huab and
Lower Ugab to the south. The pile of sedimentary rocks wedges out northward to the Unihab Basin
where Etendeka lava rests directly on the bedrock. Numerous remnants of glaciogenic rocks and
features are directly resting on the bedrock in the Huab Basin (transect 5 on Fig. 3), whereas none is
mapped at the interface between the lavas and the bedrock further north (at the border between the
Uniab and Hoanib basins, transect 5 on fig. 3). Based on these observations, we suggest that this
uneven bedrock topography covered by glaciogenic sediments corresponds to an exhumed landscape,
glaciogenic in origin. Given its dimension, i.e. at least 150 km wide and 1000 m deep, we interpreted
this glacial topographic depression that form the Huab and Lower Ugab basins (Figs. 2 and 3), as a
glacial overdeepening or trough *sensu* Benn & Evans (2010). The deepest parts of this large glacially-
carved depression were filled by sediments of the lower Karoo Supergroup (Holzförster et al., 2000),
which probably promoted preservation of the glaciogenic sediments and landforms. Shallower parts to
the north remained protruding until the outpouring of the Etendeka lavas some 165 Myr after the
glaciation, which likely obliterated all evidence for glacial activity. Furthermore, Andrews et al.
(2019) argued that profiled, elongated hills of the middle Ugab basin are glacial megalineations and
megawhalebacks indicative of paleo ice streams (Fig. 2; see also Heron et al., 2022, 2024). Few
studies have tackled the upper Ugab river valley south of the Otavi Range (Fig. 3), characterized by an





inner V-shaped gorge in which the modern river flows, at the bottom of a more subdued U-shaped

valley with a very steep northern flank and gently-sloping southern flank. The infill of Cenozoic

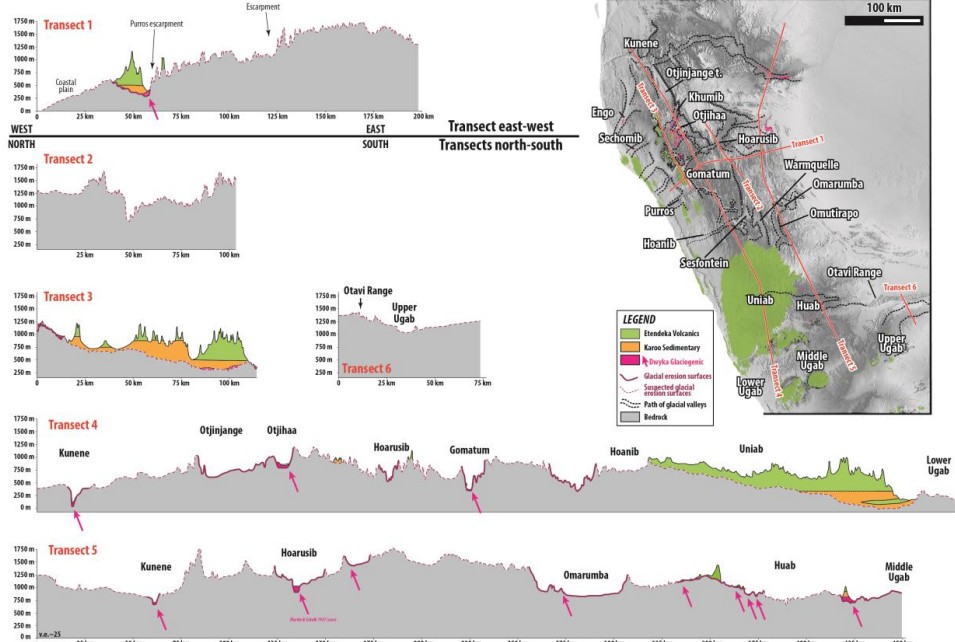

Fig. 3: Geological map indicating the Karoo Supergroup and morphostratigraphic transects across the Kaoko highland, highlighting the morphology of the Kunene, Kaoko, Huab-Ugab regions and the associated glacial valleys and troughs. Etendeka lavas are represented in green, non-glaciogenic Karoo sediments in yellow and Dwyka glaciogenics in pink, or indicated by pink arrows. Black dashed lines on the map represent outlines of exhumed glacial reliefs and valleys; solid purple lines on morphostratigraphic transects represent glacial surfaces and dashed purple lines represent suspected glacial surfaces. Bedrock in grey indicates substrate older than the Karoo Supergroup Note that this colouration is consistently used throughout the manuscript. Fig. 1b for location.

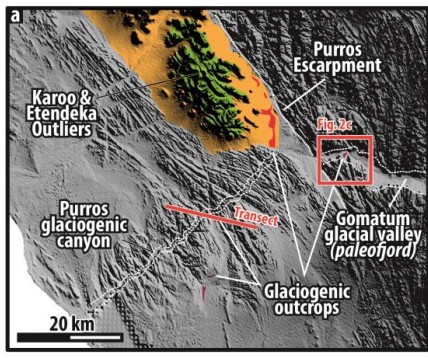

Fig. 4: (a) DEM of the western coastal Kaoko region and (b) morphostratigraphic transect. In the Purros canyon are remnants of glaciogenic sediments, and therefore the canyon is tentatively interpreted here as a relict glacial landform. See Fig. 2 for location.




sediments and the absence of Paleozoic strata as well as Mesozoic weathering processes (the northern flank of the U) led Mabbutt (1951) to posit that 'the pre-Karroo planation was not advanced'. However, the general shape (degraded U-shaped) of the valley might suggest a degraded glacial origin latter reused by Cenozoic deposition processes. In such a case, the Otavi range already existed by LPIA times. Further studies are required to confirm such a hypothesis.

## 3.2 The Windhoek Highland

The Windhoek highland lies at the center of Namibia and almost reaches 2000 m above sea level (Fig. 4). The substrate is here made of the Neoproterozoic Damara Fold Belt (Begg et al., 2009). Numerous valleys dissect and radiate outward from the Windhoek highland. Most of these valleys have been interpreted or inferred as exhumed glacial valley by Martin (1975, 1981, see also Miller, 2011). However, the only irrefutable evidence for a glacial origin is the N-S-oriented Black Nossob River valley which has a U-shaped cross-profile, 7-km wide and 30-m deep, at the bottom of which are remnants of glaciogenic sediments preserved, as indicated by geological maps (transect 2 on Fig. 4). The other river valleys dissecting the Windhoek highland, the upper and lower Swakop and its tributary, the Okahandja-Windhoek and the Kurikaub, as well as southward-flowing Skaap, Usip and Nausgamab and the westward-flowing Kuiseb have also been interpreted as glacial in origin by Martin (1975, 1981). Although these valleys have conspicuous U-shaped cross-profile (Fig. 4c), no glaciogenic sediments or morphologies have been mapped or described in the literature or observed during our own fieldwork within the thalweg of these valleys. A Dwyka outcrop has been described in the vicinity of the Nausgamab valley (Fig. 4, Faupel, 1974), which would witness at least local glacial processes; we however did not locate these deposits during own field campaign. The Okahandja-Usip is sometimes interpreted as a graben tied to Mesozoic-Cenozoic extensive processes (the Windhoek graben, Schneider, 2004; Waren et al., 2023) whereas the other valleys seem to follow grain of the underlying basement: N-S-oriented valleys follow a network of faults whilst NE-SW oriented valleys follow lithological boundaries. Further field studies are therefore required to confirm a glacial origin of the network of valleys dissecting the Windhoek Highland. Further south, Korn and Martin (1959) and Martin (1959, 1961) indicate that the encased U-shaped, E-W-oriented Tsondab valley that crosscuts the Naukluft Mountains is a glacial valley, as remnants of Dwyka sediments occur within the valley thalweg (Fig. 5d, e & f). Geological map also indicates glaciogenic sediments on the interfluve of the valley (Fig. 5d). We therefore suggest that the modern Naukluft is the southern extension of the Windhoek Highland in which glacial valleys are carved. The very name of the Naukluft which means 'narrow gorge' in Namibian German is literally after the presence of glacial valleys.



Fig. 5: (a) DEM of the Windhoek highland (central Namibia), (b) their morphostratigraphic transects. And (c) mosaic picture of the U-shaped Nausgamab valley interpreted by Martin (1961) as a potential glacial valley (see also Miller, 2011). Faupel (1974) reported glaciogenic sediments in the vicinity of this valley; (d) DEM of the Naukluft mountain crosscut by the U-shaped Tsondab valley interpreted by Korn & Martin (1959) and Martin (1961) as a glacial valley. (e) morphostratigraphic transect and (f) picture of the Tsondab valley. Fig. 1b for location.





### 3.3. The Cargonian Highland

In South Africa, between the MKB and the Kalahari Basin (Fig. 1), large areas of the Archean to Paleoproterozoic Kaapvaal Craton correspond to exhumed glacial landscapes (Fig. 5, 6 & 7). Portions of the Ghaap plateau and the Kaap-Orange river valleys (Fig. 6) as well as the Highveld, Witbank, Bushveld and Mooi-Harts areas in the Johannesburg-Pretoria region and the Vredefort Dome (Fig. 7) are part of the extensive Cargonian paleohighland. Cargonian stands for the contraction between

Carboniferous and Gondwanian (Visser, 1987a). The Buffalo-Tugela river valleys, at the southeasternmost edge of the craton, also exhibits widespread glacial landscapes (Fig. 8). Over these areas, vast relict planation surfaces, U-shaped valleys, fjords, inlets, embayments and troughs, eskers and canyons were carved by direct glacial action (Visser, 1983b, 1985, 1987a, 1997; von Brunn, 1983, 1994, 1996, Haldorsen et al., 2001; Dietrich & Hofmann, 2019). The preservation of these glacial

landscapes spans a large range from poorly-preserved to outstandingly-exposed, whose review based on geological maps, literature and own field investigations is provided below.

The most extensive relict glacial relief occurs in the confluence region between the Orange and Vaal river valleys (Fig. 6). Before joining the Vaal, the Harts River flows in a 20-30 km wide and 280-km long, NNE-SSW-oriented valley, the Kaap valley. DEM and geological maps reveal that the

valley cross profile, roughly U-shaped, is asymmetric (transect 2 on fig. 6). The eastern flank of the valley has a shallow slope (1-2%) and is formed by a ridge made up of Archean andesite of the Ventersdorp Supergroup, leading eastward to an uneven relief upon which remnants of Karoo sediments rest. Glaciogenic sediments rest in the valley axis, drape the valley flanks and, on the eastern bank, occur as pockets in paleotopographic lows that develop on the bedrock (Visser and

Loock, 1988). Here, the Nooitgedacht glacial pavement records a WSW glacial movement (Fig. 6; Slater et al., 1932; Du Toit, 1954; Visser and Loock, 1988; Master, 2012). The western flank of the valley is steep (up to 11%), 100-200 m-high and cut into Paleoproterozoic dolomites of the Griqualand West Basin forming the karstified Ghaap plateau, over which no glaciogenic sediments are mapped. The Kaap valley has therefore been interpreted as a relict paleotopography by Visser, (1987a), namely

an exhumed glacial valley carved at the interface between the Griqualand West and the Ventersdorp basins, and flowing southward from the Cargonian paleohighland. As such, this exhumed glacial valley echoes the similar, although still covered by Karoo sediment, Virginia valley inferred further east (Visser and Kingsley, 1982; Visser, 1987a, 1987b). The Hotazel valley (Fig. 6) and the valleys flowing northwestward from the Cargonian Highland toward the Kalahari basin are also interpreted as

relict glacial valleys (Visser, 1987a, 1987b, 1997). The uneven relief east of the Kaap valley onto



which glacial pavements developed and glaciogenic sediments occur is also interpreted as a relict, exhumed glacial planation surface. On the contrary, in the absence of glaciogenic sediments on the

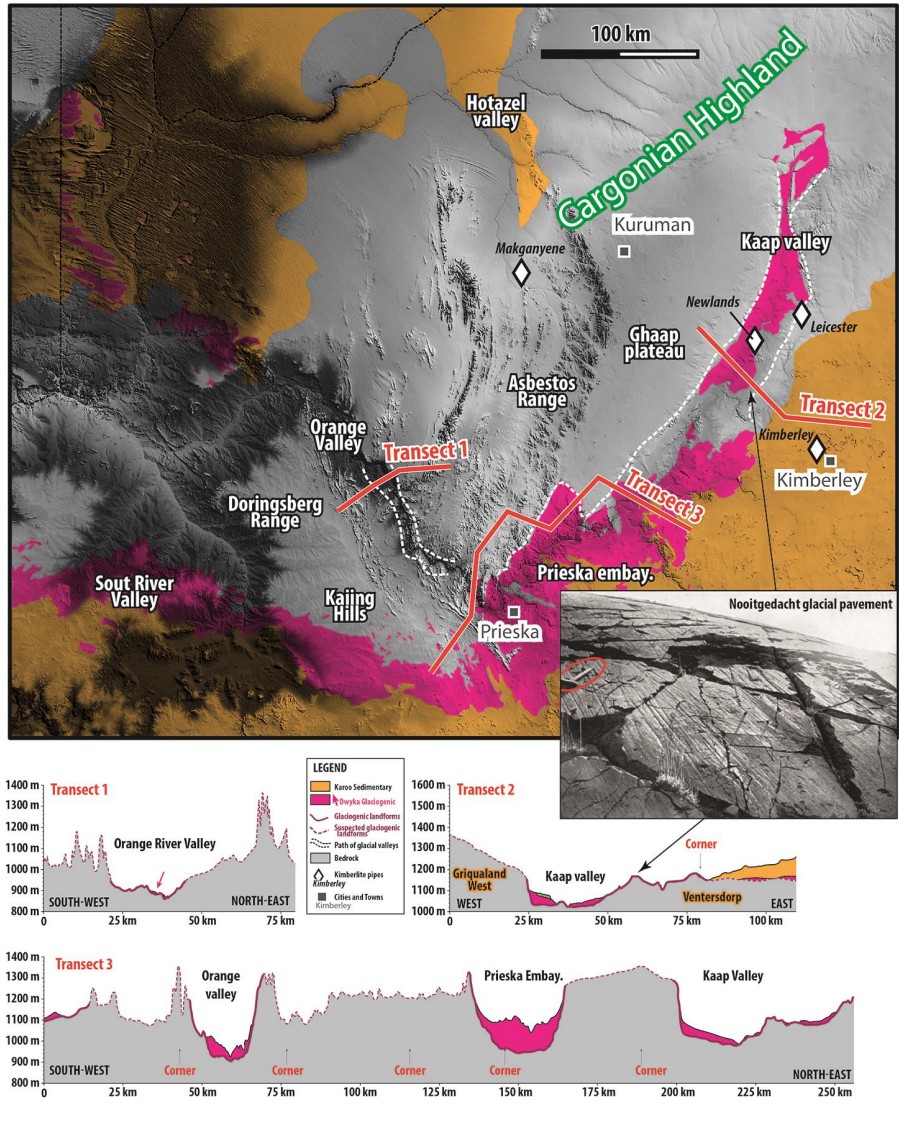

Fig. 6: DEM of the SW Cargonian Highland (central South Africa and southern Botswana; Fig. 1b for location) and associated geological transects. Widespread Dwyka outcrops in the Kaap valley visible in the landscape interpreted here as an exhumed glacial valley. Diamonds represent kimberlite pipes used for reconstruction in fig. 11. Inset photo: close-up view of the Nooitgedacht glacial pavement (whaleback) in Slater (1932). Circled hammer on the left for scale.



Ghaap plateau, it remains unclear whether this surface corresponds to a pristine glacial planation
surface or if has been reworked since (see discussion in De Wit, 2016). Further south, the Prieska
embayment is a topographic depression formed between promontories of the Ghaap plateau delineated
by steeply dipping (up to 23%), 300-m high escarpments against which the 5-120 m-thick glaciogenic
Dwyka Group onlaps and pinches out. The Prieska embayment is interpreted as a relict embayment or
glacial overdeepening (see Visser, 1987b) which formed a depocenter for accumulation of the Dwyka
glaciogenics (Visser, 1982, 1985). The Orange River itself, downstream the town of Prieska, flows in
a valley we interpret as glacial in origin: whilst remnants of glaciogenic sediments occur in the valley
thalweg, the surrounding bedrock peaks of the Doringsberg and Asbestos ranges tower some 400 m
above (transect 1 on fig. 6). This indicates that modern Orange River drainage follows an ancient
glacial trough rejuvenated by the removal of soft Karoo sedimentary rocks. The Doringsberg and
Asbestos range also already existed per se by LPIA times, and had *at least* the height they have today,
and could then have formed nunataqs if the ice was at some point during the LPIA thinner that the
height of these peaks, e.g. during recessional phases. Further west, Visser (1985) indicates that the
northwesternmost edge of the MKB consists of a succession of glacially carved basins, valleys and
embayments, such as the Sout River Valley and the Namaqua Basin, as well as promontories, ridges
and spurs, like the Kaiing hills, the Poffader Ridge and the Langberg mountains (Fig. 6). Finally, at the
westernmost end of South Africa, south of the Orange River, in the Richtersveld region, which might
correspond to the westward continuation of the Karasburg Basin, remnants of a N-S-oriented glacial
valley have been described (Fig. 1, Reid, 2015).





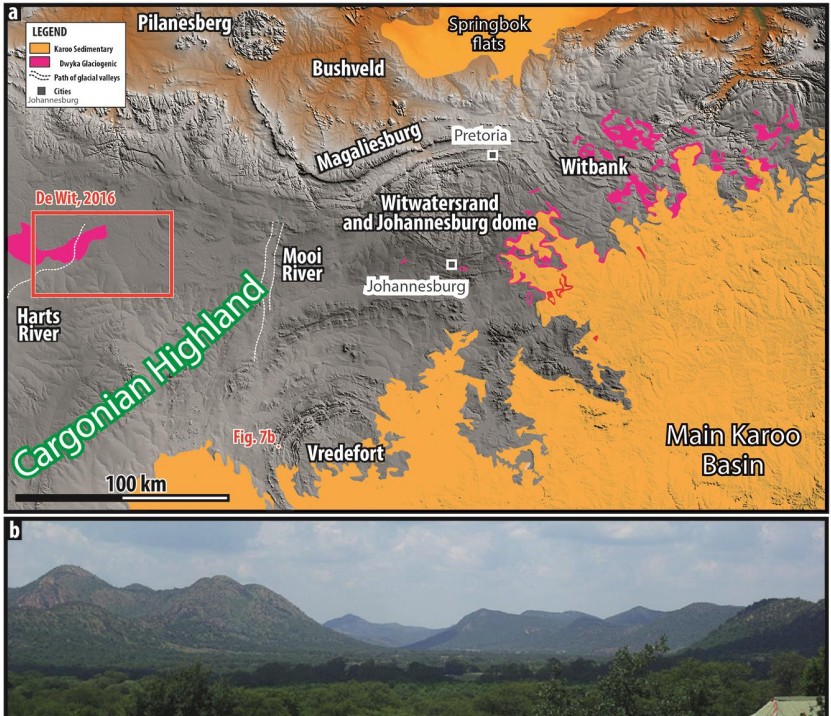

Fig. 7: (a) DEM of the central Cargonian Highland (Johannesburg-Pretoria-Witwatersrand area on the Kaapvaal craton, central South Africa). Fig. 1b for location. The Mooi and Harts river valleys, highlighted by white dashed lines, and the surrounding areas, are interpreted by De Wit (2016) as an exhumed glacial surface. Similarly, the Witbank region to the east, and the Vredefort dome to the south are also interpreted as exhumed glacial surfaces (see text for detail). In between, the Witwatersrand region, the Magaliesburg range, the Pilanesberg dome and the cities of Johannesburg and Pretoria also probably sit on a glacial surface, although further work need to be done to confirm such a hypothesis. (b) View of the Vredefort dome area where the Vaal river valley shows a U-shaped profile reminiscent of glacial erosion. A small portion of the Vaal River floodplain is seen at centre-right (Fig. 4.5 in Gibson & Reimold, 2015)

The area surrounding Johannesburg and Pretoria, including Archean basement of the Johannesburg Dome, Archean-Paleoproterozoic strata of the Witwatersrand and Transvaal supergroups forming the forming the Witwatersrand and Magaliesberg mountain ranges, southern part of the Paleoproterozoic Bushveld igneous province and the Mesoproterozoic Pilanesberg alkaline ring complex (Fig. 6), is thought to correspond to an exhumed glacial landscape (Wellington, 1937). In this

region, direct evidence for glacial processes occur on the interfluve between the Harts and Mooi River valleys. This even surface is characterized by numerous striated glacial pavements interpreted as a surface of glacial erosion, covered in places by sinuous, diamond-bearing sediment ribbons interpreted as eskers (De Wit, 2016; Fig. 7). The Harts and Mooi river valleys, incised within gently-, southward-sloping Transvaal Supergroup dolomite, have U-shaped cross-profiles and the Harts River corresponds

to the northward extension of the aforementioned Virginia glacial valley. East of the city of Johannesburg, the Witbank coal field also constitutes a pre-Karoo glacial irregular topography. In this



region, coal seams of the postglacial Vryheid Formation (Ecca Group) either rest conformably on glaciogenic deposits of the Dwyka Group and fill local hollows and depressions, 10-60 m deep, of the paleotopography inherited from glacial erosion, or directly lies on the bedrock on paleohighs (Le

Blanc Smith and Eriksson, 1979; Le Blanc Smith, 1980; Cairncross and Cadle, 1988; Holland et al., 1989; Götz et al., 2018). The mining of coal seams rejuvenates the pre-Karoo topography. Further work may reveal that this assemblage of paleotopographic highs and lows may correspond to large-scale *roches moutonnées* or even crag-and-tails similar to those encountered in Canada or Scotland and tied to the Quaternary glacial epoch. Geological maps indicate that small patches of Dwyka

deposits also occur in the Johannesburg region. Less direct morphological pieces of evidence suggest that encased ravines and canyons that dissect the cuestas formed by the Magaliesberg mountain range correspond to subglacial canyons carved during LPIA times, as suggested by Wellington (1937). Similarly, Cawthorn et al. (2015) suggested that the Pilanesberg complex forming a 100-500 m high, near-perfect circle of concentric rings of hills surrounding flat terrains of the Bushveld complex has

gained its surficial morphology and drainage pattern by the scouring of glacial ice during the LPIA. Here, however, no direct evidence for glacial action (striated pavements, glaciogenic sediments) was found.

         Further south, the Vredefort dome, the central basement uplift of a 2.1 Ga-old impact structure, displays numerous remnants of glacial erosion processes such as striae, grooves and profiled

hills, and patches of glaciogenic sediments as well as far-travelled boulders. The upper Vaal River which crosscuts parts of the impact structure has a U-shaped profile which can be interpreted as the remnant of a glacial valley (Fig. 7b). Together, this indicates that the modern landscape of the Vredefort dome corresponds to a fossil, pre-Karoo glacial landscape (King, 1951; von Gottberg, 1970a; Gibson and Reimold, 2015).

At the easternmost edge of the Main Karoo Basin (Fig. 1 & 8), the removal of less-resistant Karoo strata rejuvenates LPIA glacial landscapes sculped into the resistant Archean granites, greenstones and quartzites (Dietrich and Hofmann, 2019). The Buffalo River valley follows an inherited 100-140 m deep glacial trough carved into Pongola Supergroup quartzites, abrupt valley flanks (30-60°) made of quartzites are draped by glaciogenic clast-rich diamictites that become

horizontal in the valley thalweg (Fig. 8b). In the intervening interfluves, the landscape of rolling hills made of Archean quartzites and greenstone belt volcanics constitute a rejuvenated glacial landscape as indicated by pockets of glaciogenic sediments in paleotopographic depressions whereas paleohighs are made of basement rocks (Fig. 8c), which occasionally display hard-bed striated pavements (Fig. 8d & 8e). As for the Witbank coal field, these hills and hollows may represent crag-and-tails or field of

large *roches moutonnées*. An exhumed U-shaped glacial trough, 800 m wide, 100 m deep and 2-km in which remnants of glaciogenic sediments occur has also been observed (Fig. 8f, Dietrich and Hofmann, 2019).



The northern margin of the Main Karoo Basin hosts glaciogenic Dwyka Group rocks which reflects a threefold segmentation with regard to the paleotopography, as emphasised by numerous authors (Visser, 1987a, 1987c; Brunn et al., 1994; Von Brunn, 1996; Haldorsen et al., 2001a; Johnson et al., 2006; Isbell et al., 2008; Tankard et al., 2009; Dietrich and Hofmann, 2019; Griffis et al., 2019a, 2021): (1) The *basement-high* facies association, deposited on the Cargonian paleohighland described above, seldom exceeding a few meters in thickness, is represented by massive to poorly stratified diamictites representing subglacial till or esker deposited on land (De Wit, 2016), (2) *The valley-fill facies association* (sometimes referred to as the Mbizane Formation), up to 300 m in thickness but characterized by rapid thickness changes, consists of an alternation between massive and stratified diamictites, sandstones and conglomerates, whose deposition was largely structurally-controlled and reflects the underlying corrugated topography such as escarpments and valley walls carved into the escarpment delineating the paleohighland. (3) *The platform-basin facies association* (Elandsvlei Formation) is recognized at the centre of the Main Karoo Basin, commonly reaching 800 m in thickness, in which four deglacial sequences, consisting of alternation between diamictite and mudstones (marine to glaciomarine) units deposited in deep glaciomarine environments (outwash fans, grounding zone wedges, meltwater plumes) represent ancient lowlands.




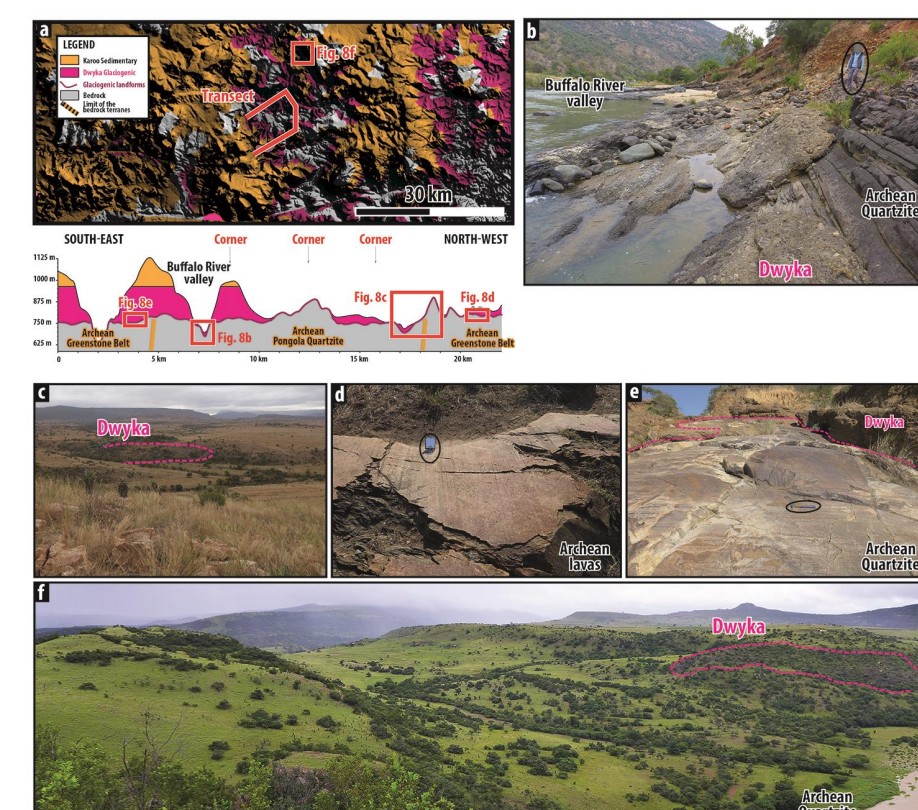

Fig. 8: (a) DEM of the eastern Cargonian Highland (edge of the Main Karoo Basin; Kaapvaal craton, eastern South Africa) and morphostratigraphic transect highlighting the Tugela valley as an exhumed glacial landscape. Fig. 1b for location. See Details in Dietrich & Hofmann (2019). (b) Bank of the Buffalo River exhuming a glacial valley. Stratified, steeply-dipping rock on the right corresponds to Archean Pongola Supergroup quartzite into which steep-flanked relief were carved and upon which coarse-grained deposits corresponding to glaciogenics of the Dwyka Group are plastered. Although the Dwyka sediments are steeply-dipping on the flank of the (paleo)valley, they become horizontal in the river thalweg. Circled geologist for scale; (c) Landscapes of rolling hills corresponding to an exhumed glacial landscape. The relief is carved into Archean Pongola quartzite seen in the foreground: topographic lows preserve remnants of glaciogenic sediments whose bulk has been eroded away by recent erosion, resurrecting the glacial landscape. Striated pavements, such as the ones showcase on fig. 7d and 7e, characterize basement floors. (d) Striated floors carved onto volcanic rocks of Archean greenstone belt, plucking of the joint at the foreground indicate an SSW ice movement. (e) Striated and polished glacial floor exposed in a stream, and showcasing a small-scale *roche moutonnée* behind the circled hammer, evidencing an ice movement to the SSW. The glacial floor is still covered in place by remnants of glaciogenic sediments. (f) A U-shaped trough, 800 m wide and 100 m deep carved by glacial erosion into Archean Pongola quartzites. Remnants of glaciogenic sediments are still present. Picture from Dietrich & Hofmann (2019).



## 3.4.   The Zimbabwe Highland

The Zimbabwe highland develops over the central region of Zimbabwe that corresponds to the Zimbabwe Craton (Fig. 1 & 9). This highland is floored by Archean greenstone belts and granites that

were intruded during the late Archean by the layered complexe of the Great Dyke (Mukasa et al., 1998). For the most part, it forms now a prominent morphological ridge that stands well above the surrounding basement-floored rocks of central Zimbabwe (Fig. 9). Over the Zimbabwe Highland, the Karoo Supergroup is represented by extremely thin (ca. 100 m) and isolated outcrops capped by 50-100 m thick Jurassic lavas of the Drakensberg Group (Rhodesia Geol. Map

1971:https://zimgeoportal.org.zw). In the Featherstone region, east of the Great Dyke, the Archean Mwanesi Greenstone Belt also forms a prominent ridge against which the Karoo sediments onlap (Fig. 9). Although no glaciogenic sedimentary series have formally been identified on geological maps ('undifferentiated Karoo') and no field study has reported glaciogenic sediments, to our knowledge, the surrounding sedimentary basins encompass evidence for glacial processes (mid-Zambezi: Bond

and Stocklmayer, 1967; Cabora Bassa: Oesterlen and Millsteed, 1994; Fernandes et al., 2023; Somabula: Moore and Moore, 2006; Tuli: Bordy and Catuneanu, 2003). We therefore posit that the Mwanesi Greenstone Belt and the Great Dyke, formed prominent reliefs during LPIA times. This relief, sealed by Karoo rocks, is now being exposed by the erosion of the sedimentary rocks and basalts. Furthermore, Moore et al. (2009) suggested that U-shaped valleys, canyons, defiles and

ravines (locally named 'poort', meaning gateway in Afrikaan) incised trough this greenstone belt as well as through the Great Dyke within which modern streams flow do not match any structural pattern and cannot have been cut by these streams. Rather, the incision corresponds to exhumed glacial valleys that are now used by streams to cross the topographic barriers (Moore et al., 2009). Assuming that the Great Dyke already existed by LPIA times implies it may have formed a nunataq when the ice

was thinner than the height of the ridge.

Further SW, in the Somabula region (Fig. 9), patches of diamond-bearing sediments attributed to Dwyka-filled hollows and topographic depressions are carved into Archean granite (Moore and Moore, 2006). The uneven topography onto which the Dwyka lies therefore corresponds to a glacial topography. We propose that the modern topography between these two localities, ca. 125 km apart,

and more generally the Zimbabwe highland made of granite and greenstone belts therefore most likely correspond to the pre-Karoo glacial landscape (Moore et al., 2009). This pre-Karoo glacial landscape should be more degraded where removal of the Karoo cover occurred earlier but at proximity of Karoo outliers, the glacial landscape must be pristine (Lister, 1987). Future field campaigns may reveal glacial features that may provide valuable clues on glacial processes and associated paleolandscapes.




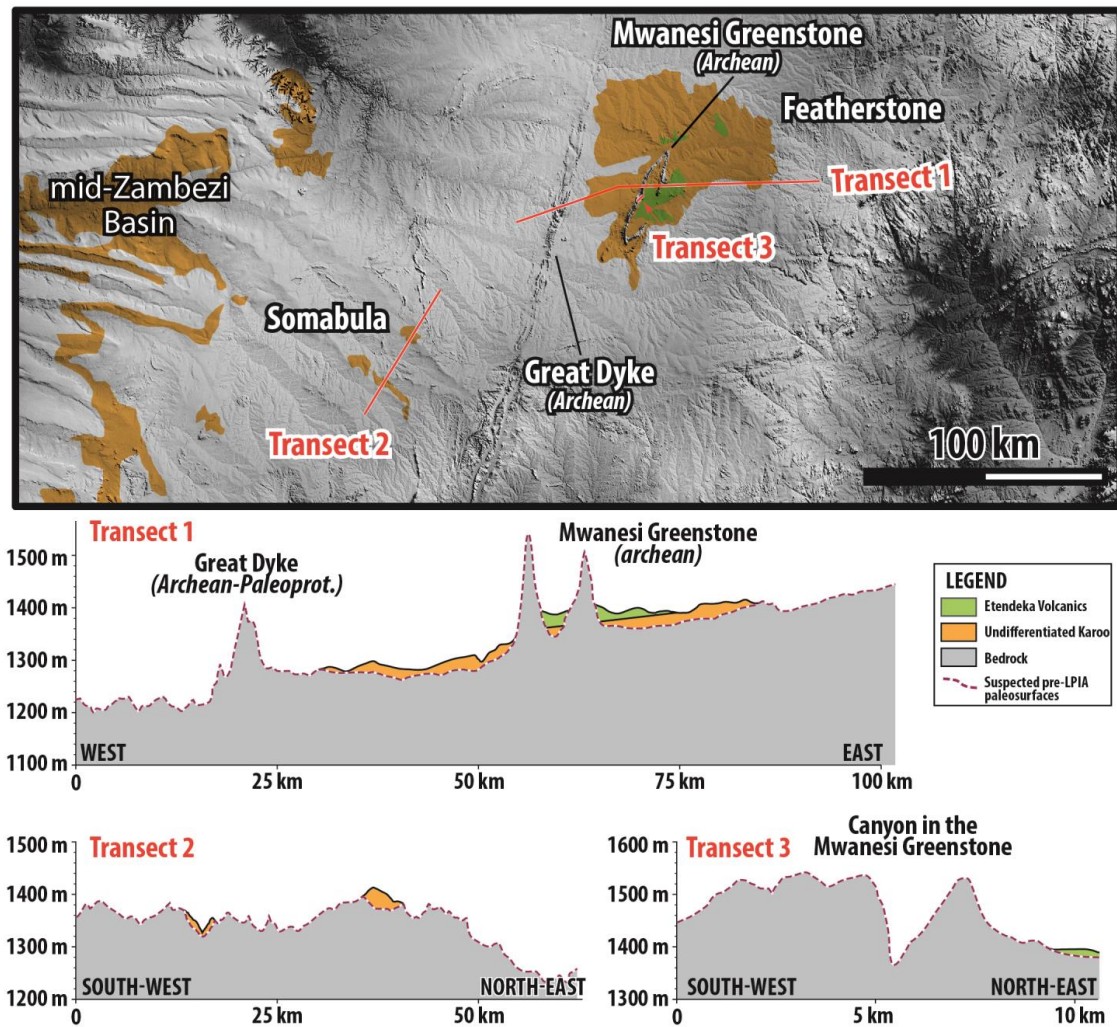

Fig. 9: DEM of the Zimbabwe Highland (central Zimbabwe) and morphostratigraphic transects across the Great Dyke, the Mwanesi Greenstone Belt and the Somabula region. The reader is redirected to Moore & Moore (2006), Moore et al. (2009) and Lister (1986) for further details. Fig. 1b for location





### 3.5. Synthesis and implications: critical analysis of the glacial paleolandscapes of Southern Africa and paleogeographic reconstruction

#### 3.5.1. The glacial paleolandscapes of Southern Africa

We have replaced and compiled the glacial paleolandscapes presented above and distinguished the indisputable from the suspected ones at the scale of Southern Africa, as presented figure 10a. The main and foremost finding deduced from our analysis is that, over Southern Africa, an area of ca. 71.000 km² consists in indisputable exhumed glacial landscapes and 360.000 km² correspond to suspected glacial landscapes, which together correspond to ca. 10% of the total area of the region (Fig. 10a). Compared to area floored by a substrate older than the Karoo Supergroup, i.e. older than ca. 300 Myr (ca. 1.700.000 km²), this proportion rises to ca. 25%, as the glacial paleolandscapes are mostly found on the paleohighlands formed by Archean and Paleoproterozoic terrains. It must be noted however that the exact delimitations and extent of the suspected glacial landscapes on figure 10a have been plotted on the basis of morphological features described above but may require local reassessments.

From that map, it appears that many prominent aspects of the modern morphology of Southern Africa in fact correspond to ancient, rejuvenated paleolandscapes. Notably, some modern escarpments are in fact exhumed paleoescarpments delineating the paleohighlands from the basins, such as in the Kaoko region of Namibia or along the Kaap Valley in South Africa. In other instance, the escarpment is still buried under Karoo sediments, such as in central South Africa and Lesotho and southern Botswana, deduced from abrupt increase in thickness in Dwyka glaciogenic sediments toward the south, as observed from drilling (Fig. 10a; see Visser, 1987a, 1987b). Also, the modern river drainage follows the pattern of inherited glacial relief: the Vaal River in the Kaap Valley and the Orange River in the Orange Valley (South Africa), the Kunene and other NW Namibian rivers in the fossil fjords as well the Ugab, Swakop and Black Nossob rivers and the Zambezi River funneled by the Zambezi escarpment (Zimbabwe). In addition, narrow ravines cut into prominent topographic barriers, such as the Great Dyke and Mwanesi in Zimbabwe, the Magaliesberg and Doringsberg-Asbestos mountains in South Africa and the Naukluft in Namibia, seem to correspond to exhumed glacial gorges. Our study therefore highlights the absolute necessity of integrating the morphological legacy of the Late Paleozoic Ice Age when assessing the evolution and modern aspect of the landscapes of Southern Africa (see below discussion section 4.1).

It must also be stressed that the exhumation of the glacial relief may provide valuable clues about paleoaltitudes and finite uplift of the African continent since the late Paleozoic, only rarely constrained and quantified (Simoes et al., 2010; Braun, 2018). Indeed, the paleofjords of Namibia bear



sedimentological evidences for coastal, and sometimes even intertidal, environments (Dietrich et al., 2021), providing valuable clues about the paleo zero altitude. Even though postglacial eustatic and isostatic processes likely modified the relative sea level at this time, by at most a few hundreds of meters (Montañez and Poulsen, 2013; Dietrich et al., 2018, 2021), the presence of coastal sediments tied to the LPIA nowadays observed at 300-400 meters above modern sea levels within the paleofjords indicate that 1. the finite uplift of Southern Africa since the Late Paleozoic was of a similar value, and 2. the interfluve of the paleofjords immediately after the LPIA stood at an altitude corresponding to *at least* the elevation difference between them and the valley thalwegs, considering that the interfluves may have been eroded and levelled down since then. Similar findings would most likely be unravelled in every glacial paleovalley of Southern Africa.

### 3.5.2. Paleogeography

Based on the map of exhumed glacial paleolandscapes (Fig. 10a), on the compilation of sedimentary facies as well as on previous local paleogeographic reconstitutions (e.g., Smith, 1984; Lister, 1987; Visser, 1983b, 1985, 1987a, 1987b, 1989, 1992, 1993, 1997; Daly et al., 1989; von Brunn, 1991, 1993; Veevers et al., 1994; Smith et al., 1993; Johnson et al., 1996, 1997; Haldorsen et al., 2001; Isbell et al., 2008; Dietrich et al., 2019a, 2021), we have attempted to reconstruct the paleogeographic configuration at the scale of Southern Africa at the end of the LPIA, as presented figure 10b. A threefold morphological pattern is evident, consisting of (1) highlands locally covered with hill and mountain ranges; (2) escarpments into which glacial valleys and fjords are incised, and that lead downstream to (3) sedimentary basins (the Karoo-aged basins) that correspond to the lowland counterparts of the highlands. It must be noted that this map is interpretative and many uncertainties remain. Notably, we propose that the original extent of the Karoo-aged sedimentary basins was greater than their modern outcrops, as offshore data indicate the presence of Karoo sediments on the modern continental margin. As the offshore Walvis and Lüderitz Basins hosts Karoo sediments (Clemson et al., 1997, 1999; Aizawa et al., 2000; Baby et al., 2018b; 2020), we have extended the Aranos and Kaoko basins further west and connected them to their offshore counterparts. Martin (1973b) states that, as no glacial erratics sourced from the Kaoko have been found in the Parana Basin, the Brazilian Karoo equivalent, a topographic depression or a basin, perhaps oceanic, should have existed between Namibia and Brazil during the LPIA, likely corresponding to the northern realm of the Walvis Basin. And he concludes that '*paleogeographic evidence does not easily fit into the concept of a direct join of the African and the South American continental plates*'. Griffis et al. (2021) moreover indicate that only Gondwanan-scale deglacial events permitted the delivery of African-sourced sediments into the Parana Basin while glacial flows between Africa and South America were hindered, suggesting the presence of substantial topographic barriers such as a basin that would have deflected/hindered ice flows. For the connection between the Aranos and Lüderitz basins and their extent further west, the





Fig. 10: (a) Synthesis of glacial paleolandscapes at the scale of Southern Africa. Dark pink indicates attested glacial surface and light pink suspected glacial surfaces whose compilation is based on the presence of glacial morphological features (see text for details). Dark grey regions are Karoo-aged basins. Exhumed paleo-escarpments are represented by black bold lines and escarpments still buried under sediments are after Visser, 1987a, 1987b. Light orange region corresponds to surficial sediments of the Kalahari Desert, after Haddon (2005). (b) Proposed paleogeographic reconstruction of Southern Africa at the end of the LPIA. Blue-grey areas represent highlands whose names are written in green, sedimentary basins are represented in dark yellow where attested or light yellow where suspected. Escarpments delineating the highlands from the basins and glacial valleys carved into it are represented as bold solid lines where attested or as dashed lines where suspected (see Visser, 1987a, 1987b). Hills or mountainous regions are also indicated. Region where no data is available mostly correspond to the Kalahari Desert – see fig. 10a above. Names of glacial valleys and escarpments refer to those discussed in the text to which the reader is redirected for further details.



absence of sediments between these offshore-onshore realms would be explained by their removal
through the post Gondwana-breakup functioning of the escarpment passive margin (see above section
2.4; Braun et al., 2014; Braun, 2018a), which nowadays delineates the western border of the Aranos
Basin. This very escarpment therefore postdates the LPIA. In line with this, Visser (1987b) states that
'*Towards the west, [the Kalahari] basin probably opened into a sea. Martin (1973b) favoured the
extension of the Kalahari basin into South America as goniatites of the same subgenus were found in
Uruguay and Namibia in very similar stratigraphic positions. Those deposits, however, formed during
an interglacial when large parts of SW Gondwana were inundated as a result of sea-level rise*'. It
must be noted that the tectonic nature of the Lüderitz and Walvis Basins, the offshore counterpart of
the Aranos and Kaoko basins, remains contentious, as they may relate to the functioning of the Parana
and Karoo basins (Pysklywec and Quintas, 1999; Pysklywec and Mitrovica, 1999; Catuneanu, 2004;
Catuneanu et al., 2005c) or to a Late Paleozoic rift system (see below discussion section 4.2; Clemson
et al., 1997, 1999; Aizawa et al., 2000).

About the offshore continuation of the Main Karoo Basin, Karoo sediments have also been found
both in the offshore Orange Basin to the west (the South Atlantic Sea Arm of Visser, 1987b, sea also
Baby et al., 2018b) and in the Durban Basin (Baby et al., 2020) and on the Falklands-Malvinas Islands
(the Dwyka-equivalent Fitzroy tillite), whose restored position is off the modern SE coast of South
Africa (Hyam and Marshall, 1997; Meadows, 1999; Stone, 2016). Here also, the removal of the
connection between the onshore and offshore basins would relate to post Gondwana-breakup history
and erosion of the plateau and removal of the sedimentary cover near the uplifted passive margins and
in the Cape Fold Belt orogeny (Wildman et al., 2016). Finally, the Karoo sediments may have
extended up to the modern coast of Mozambique, as Karoo sediments crop out at the South Africa-
Mozambique border, dipping west (Viljoen, 2015) and Karoo sediments and volcanics are observed on
seismic imagery at the base of the Limpopo and Zambezi coastal basins (Salman and Abdula, 1995;
Ponte et al., 2019; Senkans et al., 2019; Roche et al., 2021; Roche and Ringenbach, 2022). As for the
Namibian margins, the tectonic nature of these coastal Mozambique Karoo-aged basins remains
unknown, and may also relate to a Late Paleozoic rift system (the Karoo I rift of Frizon de Lamotte et
al., 2015).

Another uncertainty concerns the nature and origin of the Upper Ugab and Waterberg valleys of
Northern Namibia, the Karasburg Basin of Southern Namibia (a glacial overdeepening for Martin,
1973b, see also Visser, 1987b) as well as the Tshipise, Tuli, Ellisras and Springbok Flats basins of
South Africa and Zimbabwe. Although some authors invoke rift processes in the genesis of these
basins (Daly et al., 1989; Smith and Swart, 2002; Frizon De Lamotte et al., 2015; Guillocheau, 2018),
the reduced thickness of the entire Karoo Supergroup (100-200 m) within these basins (Holzförster et
al., 2000; Smith and Swart, 2002; Bordy and Catuneanu, 2003; Johnson et al., 2006; Bordy, 2018),



indicating very low accommodation and subsidence, and in some cases the absence of faults bordering
these Karoo strata question such an interpretation.

        In spite of the uncertainties, our reconstruction may serve as the basis to provide clues into the
dynamics of the associated ice sheets controlled by local-regional topography. In such a configuration,
the highlands would correspond to ice divides and the ice would have been drained through the
escarpments toward the basins and carved the valleys which in turn further promoted the funneling of
the ice (for example, see fig. 5 in Smith et al., 1993; fig. 4 in Isbell et al., 2008; Griffis et al., 2021).
Also, the presence of topographic escarpments may have locally acted as pinning point for ice margins
during periods of ice retreat (Haldorsen et al., 2001; Dietrich et al., 2017; Lajeunesse et al., 2018).
Importantly, the presence of vast and well-preserved glacial paleoreliefs along with their related
glaciogenic sedimentary deposits in the basins, as showcased on Fig. 10b, make Southern Africa the
ideal place to establish the first-ever quantification of source to sink budgets through a complete
icehouse cycle, over tens of millions of years. Such a finding would be indispensable to include long-
term glacial erosion and sedimentation processes within global assessment of landscape evolution and
sediment fluxes through hundreds of millions of years (Salles et al., 2023). In turn, this glacial erosion
and the resulting production and export of sediment may have acted as a significant long-term carbon
sink through weathering of the newly eroded substrate and sediments and burial within glacially-
carved reliefs (Smith et al., 2015; Cui et al., 2016, 2022) that may explain secular climate change
during the LPIA (Montañez et al., 2016; Myers, 2016; Goddéris et al., 2017).

## 4. From LPIA to present: burial and exhumation history, preservation and rejuvenation of glacial landscapes


        The preservation through geological times of the fossil glacial landscape -the paleoreliefs-
described above and their present-day exposure (Fig. 10a) require that they have been buried under
younger sediments or lava flows after their carving and later exhumed and stripped off their infill,
owing to strong erodibility contrast between their substrate (resistant crystalline and metamorphic
rocks) and infill (erodible and weatherable volcano-sedimentary rocks). In the following, we provide
burial-exhumation histories of these glacial landscapes (Fig. 11) on the basis of local/regional
sedimentological, stratigraphic, magmatic, geomorphologic and thermochronometrical data as well as
other available information for constraining uplift and erosion, such as the location, ages and
exhumation history of kimberlite pipes and erosion-deposition budgets (see section 2.4). The burial-
exhumation history is given for the Kaoko paleohighland (Fig. 11a), the southern margin of the
Cargonian paleohighland (Fig. 11b) and the Zimbabwe paleohighland (Fig. 11c). For the need of the
reconstruction of the burial-exhumation history from thermochronometrical data (apatite and zircon



fission tracks, (U-Th-Sm)/He on apatite), geothermal gradients of 25°C.km$^{-1}$ are assumed for the Kaoko, and Zimbabwe and Cargonian highlands (Mackintosh et al., 2019; Macgregor et al., 2020). As

an example, a warming/cooling of 100°C would indicate a burial/exhumation of 4 km. The history proposed here spans the whole period between the LPIA (ca. 300 Ma) and today. Given the discrepancies in data availability between these three regions, the level of details is significantly





Fig. 11: Burial-exhumation history models the Kaoko (Fig. 2), Cargonian (Fig. 6) and Zimbabwe (Fig. 9) highlands. Thermochronological inferences are provided in the graphs, exhumation evidenced from kimberlites for the Cargonian Highlands are displayed in red and sediment volume accumulated on the continental margins are showcased in yellow. Raab et al. (2005), Krob et al. (2019) and Margirier et al. (2019) for the Kaoko; Stanley et al. (2015, 2019, 2021) and Wildman et al., 2015 for central South Africa and Mackintosh et al. (2017) for central Zimbabwe.



different and the stages/ages highlighted may not be equivalent. Finally, we would like to stress that assessing the controversial exhumation history of this region is beyond the scope of the paper and we objectively provide information we have at hand.

## 4.1. The Kaoko highland

The burial-exhumation history of this region (Fig. 11a) is provided on the basis of extensive
sedimentological and geomorphologic findings (Figs. 3 and 4) as well as thermochronometrical constraints. Margirier et al. (2019) was used since the early Cretaceous, where Raab et al. (2005) and Krob et al. (2020) were used since the LPIA, although the geological set-up in Krob et al. (2020) may be too restrictive.

### 4.1.1. Burial history

In the Kaoko region, the LPIA is represented by glaciogenic landforms and by thin, less than 20 m-thick glaciogenic sedimentary rocks confined within the paleofjords. From the demise of the LPIA until Early Jurassic (190 Ma), i.e. for 110 Ma, thermochronological data indicate a warming of ca. 35°C (Krob et al., 2020), i.e. a burial of 1.4 km considering the thermic gradient described before. This burial corresponds to the deposition of the lower Karoo Supergroup. At this stage, remnants of
pre-Karoo glacial topography would be largely buried. Depositional environments associated to this accumulation are largely unknown as few remnants of these strata remain, apart from the lowermost succession ("undifferentiated Karoo" Fig. 2), immediately overlying the glaciogenics, consisting of marine and deltaic facies. Clues about detailed timing and depositional environments may arise from the neighbouring Huab Basin (Fig. 1) where the lower Karoo Supergroup consists in fluvial
(Verbrandeberg Fm), deltaic (Tasrabis Fm), shallow marine (Huab Fm), lacustrine (Gai-As Fm) and fluvial (Doros Fm) sediments of lower to middle Permian (Holzförster et al., 2000) that onlap on LPIA paleo-topography (Erlank et al., 1984).

    Thereafter, between 190 and 150 Ma, a cooling of 25°C is indicated by Krob et al. (2020), corresponding to a denudation of 1 km. This erosion renewed the pre-Karoo landscape for the first
time. This is attested by the presence of alluvial (formation not named, Schreiber, 2011), locally aeolian (Twyfelfontein Fm.) and particularly colluvial sediments, encompassing locally-derived, highly immature clasts, that are present in the valleys, well below the interfluves. Meandering rivers flowed in the valley axis whilst colluvial aprons, similar to the modern ones (Fig. 2c), abutted against valley walls that provided materials. These alluvial/colluvial sediments form the Upper Karoo
Supergroup that unconformably cover the Lower Karoo Supergroup (Jerram et al., 1999, 2000). The origin of this denudation event is poorly understood but might be related to regional uplift prior to the Etendeka volcanic eruption. Sediments forming the Upper Karoo Supergroup are themselves



underlying, and sometimes interdigitated with, the Etendeka lavas, indicating an age of 135 Ma. In total, the Karoo Supergroup in the Kaoko region is 250-350 m thick and spanned 165 Ma, from ca. 300 to 135 Ma.

The outpouring of the up to 2.8 km thick Etendeka volcanics at 135-132 Ma, part of the Paraná-Etendeka Large Igneous Province (Dodd et al., 2015; Gomes and Vasconcelos, 2021) covered and sealed the Karoo sediments and the remaining pre-Karoo topography, against which the lavas onlapped. Maximum thickness was therefore likely emplaced in topographic depression whereas highs were covered in thinner lavas (Margirier et al., 2019).

### 4.1.2. Exhumation history

Thermochronological data along the Kaoko Belt indicate a progressive cooling of 160°C from 130 Myr to today (Krob et al., 2020). Partly contradictory thermochronological data from Margirier et al. (2019) indicate a two-step cooling history over the same period, reaching about 290°C of cooling. According to these authors, the period between 135 and 100 Myr is characterized by a cooling of ca. 250°C, interpreted as ca. 900 m of erosion coupled to a decrease in the geothermal gradient due to magmatic cooling after the Etendeka LIP eruption. Margirier et al. (2019) then indicate that between 100 and 65 Ma, little to virtually no denudation occurred while Raab et al. (2005) indicate major exhumation of 1.5 km between 80 and 60 Ma. Denudation then resumed, by ca. 1.6 km of erosion (40°C of cooling), until 35 Ma, when most removal of the highly weatherable Etendeka lavas occurred, most likely enhanced by a humid and warm climate as suggested by the abundant detrital kaolinite in well data from the offshore Walvis Basin (Holtar,. Forsberg, 2000). Siliciclastic volumes preserved in the Walvis Basin show an increase in sedimentation during the Upper Cretaceous, coinciding with the first pulse of denudation revealed by thermochronological data. The acceleration of the denudation during the Paleogene highlighted by Margirier et al. (2019) is not detected in the basin. The humid climatic conditions at this time in southern Africa (Braun et al., 2014) can explain this contradiction by enhancing the chemical erosion. From 35 Myr to the present-day, cooling was virtually non-existent, suggesting limited erosion in the region, which has allowed the preservation of the pre-Karoo glacial landforms. This is consistent with the climate aridification of the region during the Middle-Late Miocene (Pickford and Senut, 1997) coinciding with the establishment of the offshore Benguela Current (Siesser, 1980; Diester-Haass et al., 1990).

## 4.2. The Cargonian Highland

The burial-exhumation history of the glacial landscapes of the Kaapvaal craton showcased here focuses on the Kaap and Orange river valleys and the adjoining Ghaap plateau and Asbestos range (Fig. 11b). This history is based on thermochronological data (Flowers and Schoene, 2010; Kounov et al., 2013; Wildman et al., 2015, 2016, 2017; Baughman and Flowers, 2020 and references therein),



dating and erosion of kimberlite pipes (Partridge, 1998; James, 2003; Hanson et al., 2009; Stanley et al., 2013, 2015, 2021) and stratigraphic inferences and dating of ash layers present throughout the Karoo Supergroup succession (Bangert et al., 1999; Johnson et al., 2006; Fildani et al., 2007, 2009;

Mckay et al., 2015; Belica et al., 2017; Griffis et al., 2018, 2021). The following burial-exhumation history until nowadays seems to be twofold in this region, characterized by an early burial (the deposition of the Karoo sediments and volcanics accumulated in the aftermath of the LPIA until 183 Ma, date of the outpouring of the Drakensberg LIP) and a late exhumation tied to the polyphase activity of the African Superplume.

4.2.1.   Burial history

The burial history of this region relates to the evolution of the Main Karoo Basin (MKB) topped by the Drakensberg Group lavas. The Karoo sediments pinch out to the north (Johnson et al., 1997), from 8-9 km in thickness at the center of the Basin to 1.5 km south of Johannesburg (Fig. 1), the modern limit of the basin being erosional.

The onset of the MKB started with the demise of the LPIA dated to 296 Myr ago (Griffis et al., 2019a, 2021). Spatial extension of the Dwyka sediments is unknown: although it accumulated in the valleys (valley-fill facies association), no evidence indicates that the Ghaap plateau and Asbestos range were covered in glaciogenics. Sediments and lava of the Karoo Supergroup then accumulated; whereas thickest accumulation occurs at the center of the MKB, little is known about the thickness

that accumulated on the paleohighlands (see figure 6 in Hanson et al., 2009). Thermochronological data are partly contradictory. Wildman et al. (2017) indicate that a linear cooling of 60°C occurred from 350 Myr to today, which would imply 2.4 km of erosion. In line with this, Hanson et al. (2009) and Stanley et al. (2013, 2015) postulate on the basis of kimberlite pipes that ca. 1.5-2 km of Karoo sediments have been eroded from the Ghaap plateau, as indicated by the hypabyssal facies of the

Makganyene kimberlite cropping out at the surface. Downrafted clasts derived from the surface at the time of kimberlite emplacement indicate that *at least* the Drakensberg basalts covered the paleohighland (Hanson et al., 2009). Contradictory to this model, Baughman and Flowers (2020) and Flowers & Schoene (2010) indicate an abrupt warming of 60°C between 280 and 250 Ma, followed by a quiescent period until 100 Ma. We posit that the whole Karoo Supergroup that existed over the

paleohighland must have been thinner than its counterpart in the basin by *at least* the difference in altitude between the paleohighland and the basin (ie the original height of the escarpment). Based on these findings, we have chosen to limit the extent of the sedimentary Karoo to the sedimentary basins whereas the Drakensberg basalts covers the entire study area, including the Ghaap plateau.

   4.2.2.   Exhumation history

The stratigraphy of the margin at the mouth of the Orange River demonstrates the presence of a proto-Orange delta from the uppermost Lower Cretaceous draining a large portion of the subcontinent (see Figure 4 in Baby et al., 2018b). This agrees with the thermochronological data from the southern



margin of the Cargonian Highland, which indicates denudation has started from 115 Myr (Flowers and Schoene, 2010), in agreement with the denudation pattern inferred from kimberlites erosion (Hanson

et al., 2009; Stanley et al., 2013; Wildman et al., 2015). On a broader scale, thermochronometric data from Stanley et al. (2021) reveal two denudation pulses coinciding with an acceleration in offshore siliciclastic sedimentation rates (Baby et al., 2020), interpreted as reflecting the growth of the South African Plateau (Braun et al., 2014). The first major pulse occurred between 90 and 70 Myr (700 to 1200 m of exhumation) and a secondary one between 30 and 10 Myr (500 m) (Stanley et al., 2021).

### 4.3.   Zimbabwe Highland

The burial-exhumation history of Zimbabwe is less well-constrained than its Namibian and South African counterparts, notably as no detailed stratigraphic surveys have been conducted. Strong discrepancies and contradiction moreover exist between geological and thermochronometrical data. The burial-history, as detailed below, is two-fold (Fig. 11c).

#### 4.3.1.   Burial history

In Zimbabwe, the precise age of the demise of the LPIA as well as glacial dynamics (ice flow directions, ice margin fluctuations etc.) and ice thicknesses are unknown. After glacial retreat, the landforms were covered by Karoo fluvial sediments of late Triassic age (Seward and Holtum, 1921) whose thickness is extremely restricted, from 0 (Somabula area, MacGregor, 1921) to 30 m

(Featherstone area, Anderson, 1978). In Zimbabwe, sedimentary equivalent of the Main Karoo Basin of South Africa is either missing or highly condensed. The overlying Karoo basalts are also very thin, 10-50 m preserved in the Featherstone area (Anderson, 1978). Thermochronological data   from Macintosh et al. (2017) indicate that a ca. 50°C warming occurred, from 300 to ca. 40-25 Ma, corresponding to a burial of 2 km. Compared to the preserved sediment thickness,

thermochronological data would imply that a an almost 2 km-thick accumulation of Karoo sediments and/or basalts have been eroded away.

#### 4.3.2.   Exhumation history

Exhumation history is provided by thermochronological data (Macintosh et al., 2017) that indicate that denudation started around 40-25 Myr most likely due to uplift of the region. This is in good agreement

with the initiation of the modern Zambezi Delta whose catchment area includes the studied area around 35 Myr (Salman and Abdula, 1995; Ponte et al., 2019). As for the Cargonian Highlands, the offshore stratigraphy of the margin surrounding Southern Africa can provides clues to the history of denudation on land. Thus, the sedimentary isopach map from Baby's (2017) (Figure 7.5 in Baby, 2017 and Figure 7.2 in Ponte, 2018), demonstrates the existence of a Limpopo proto-delta whose watershed

may have drained the Zimbabwe region as early as the Lower Cretaceous.





# 5. Implication and Discussion

## 5.1. Post-LPIA evolution: rejuvenation of glacial surfaces, implication for paleoaltitudes and finite uplift of Southern Africa

We have shown that high-standing plateaus of southern Africa floored by Archean and Proterozoic rocks are exhumed glacial landscapes dating back *at least* 300 Ma, and therefore relate to the Gondwanan surfaces of King (1948), Twidale (1998) and others (e.g. Aizawa et al., 2000). Glacial landforms define a 'surface of glacial landscape' that used to be covered by thin Karoo sediments whose lowermost unit is the glaciogenic Dwyka Group deposited during the Late Paleozoic Ice Age (LPIA). Accordingly, this surface has been termed 'ancestral glacial pre-Karoo peneplain' (Wellington, 1937), 'sub-Karoo surface' (King, 1948) or more specifically 'pre-Dwyka topography' (du Toit, 1954; von Gottberg, 1970). Therefore, this 'pre-Dwyka topography' was carved through glacial scouring during the LPIA.

The preservation and latter rejuvenation of these glacial paleolandscapes are tied to a complex burial and exhumation history. It has been stated by Lister (1987) that these relict glacial landforms are '*most accurately seen in proximity to their contact with the cover*'. Therefore, these relict glacial landsurfaces are preserved and cropping out owing to the combination of three parameters:

(i)  The glacial landsurfaces were sufficiently covered by (Karoo) sediments that protected them from erosion and further obliteration.

(ii)  The sedimentary piles that once covered these surfaces were, on the other hand, thin enough, on paleo-upland and/or area characterized by weak subsidence, to have been eroded away by post-LPIA erosion, in order to expose the relict surfaces (Fig. 1b). These areas of limited sedimentary accumulation and rejuvenated glacial surface contrast with the center of the Karoo-aged basins where the sedimentary piles are too thick to have been eroded away in recent times. At the other end of the spectrum, areas that experienced early exhumation, because having been covered by only thin sediments, not covered at all, or experienced early tectonic uplift, have been eroded away and overprinted by more recent erosion processes. Lister (1987) summarized this concept as '*Older landsurfaces [...] are thereby buried or fossilized until such time as the overlying sediments or lavas are removed, thus permitting the older landsurfaces to become subaerial once again. Modern erosion quickly destroys the resurrected landsurfaces so that their original form is most accurately seen in proximity to their contact with the cover*'

(iii) The erodibility contrast between the weathering-resistant Archean to Proterozic basement into which the glacial reliefs developed, and the weaker, prone to erosion sedimentary and volcanic cover likely played a significant role in rejuvenating these surfaces (see also Braun et



al., 2014). Accordingly, post-LPIA erosion was likely significantly slowed down when reaching the basement that therefore acted as a structurally-controlled erosion surface.

Therefore, these surfaces characterized by glacial landforms call into question their previous interpretation as pediment (flat erosional surfaces bounded by escarpment but eroded in arid or humid condition *sensu* Guillocheau et al., 2018) tied to the activity of the African superplume that led to

stepwise uplift of southern Africa since the Mesozoic (Braun et al., 2014; Guillocheau et al., 2018; Baby et al., 2020). These paleolandscapes were characterized by stepped surfaces, escarpments and valleys which cannot therefore be interpreted as planation surfaces and pediments (Guillocheau et al., 2018). Care must therefore be taken when using the glacial surfaces as proxies for uplift of southern Africa or when budgeting Mesozoic-Cenozoic erosion-sedimentation between land and continental

margins (Baby et al., 2020). Our study therefore emphasis the absolute need of considering the morphological legacy of the Late Paleozoic Ice Age when assessing the evolution and modern aspect of the landscapes of Southern Africa.

### 5.2.    Pre-LPIA evolution: existing reliefs amplified by glacial erosion?

Three morphological segments are delineated within the 'pre-Dwyka topography' formed by glacial

erosion and planation processes during the LPIA, namely (1) the highlands, (2) the escarpments into which glacial valleys and fjords are incised, and (3) the sedimentary basins (Fig. 10, 11 & 12). An array of sedimentological, thermochronometrical, structural and chronological data however suggest that glacial planation processes may have in fact reshaped and/or amplified an even older landsurface that existed before LPIA times, as it has been emphasized by various authors over the last century.

Below is a discussion on the origin of this threefold segmentation with regard to crustal and basement structures, tectonic history and pre-LPIA erosion processes. As detailed below, the pre-Dwyka glacial landscape likely resulted from a combination of ice sheet dynamics, pre-glacial landscape evolution and underlying geology just as it is the case for Quaternary glacial landscapes (Jess et al., 2019).

#### 5.2.1.  Highlands: planation surfaces, peneplains or rift shoulders reshaped by glacial

processes?

Over the Cargonian highland of the Kaapvaal craton, thermochronometrical data (Baughman and Flowers, 2020) indicate that cooling, which reflects exhumation and erosion, prevailed since 1 Ga until the LPIA, suggesting that the LPIA was the ultimate episode of a 700 Ma-long history of erosion and planation. Furthermore, Archean rocks at the center of the Kaapvaal Craton yield apatite fission

track (AFT) ages of 331.0 ± 11.0 Myr and 379.0 ± 23.0 Myr (Wildman et al., 2017), In line with this, De Wit (2016) suggested that the pre-Karoo surface is a palimpsest that corresponds in fact to an older surface that already developed before the deposition of the Pretoria Group (Transvaal Supergroup)



during the Paleoproterozoic, was rejuvenated in pre-Karoo times by the scouring action of glaciers, and nowadays again cropping out (Eriksson et al., 2001).


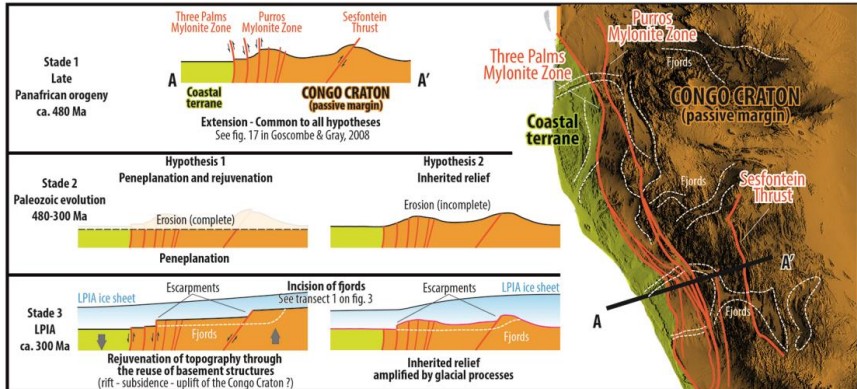



Fig. 12: Structural map of the Kaoko Highland (Figs 2 and 3), faults are after Goscombe & Gray (2008) and two alternative models for the evolution of the escarpments and the associated valleys before the LPIA, as follows: Hypothesis 1 implies that the relief created at the end of the Panafrican orogeny around 480 Myr was entirely levelled down (peneplanation) before the LPIA and rejuvenated owing to vertical crustal movements during or immediately prior the LPIA whose ice flow carved valleys into it. Hypothesis 2 implies that the relief created by the Panafrican orogeny was only partly eroded during

the time interval between the Panafrican orogeny and the LPIA and was amplified by glacial processes. The first stage representing the end of the Panafrican orogeny – extension tied to post-orogenic collapse – is common to both hypothesis and derived from the Goscombe & Gray (2008). See text for details.

The Kaoko region of NW Namibia corresponds tectonically to the Kaoko Branch of the

Panafrican orogen that developed between the Congo Craton and other terranes 580-480 Myr ago (Goscombe and Gray, 2008). Therefore, the region may have been flattened through >180 Myr of peneplanation that had followed the orogeny until the LPIA. In that case, the Late Paleozoic highland would originate from the uplift of this peneplain immediately prior to, or during the LPIA. An alternative hypothesis would be that the Kaoko highland had remained high since the Cambrian

(Doucouré and de Wit, 2003). This would require that the Kaoko relief was inherited from and compensated by the crustal structure of the Panafrican Orogen. Such a hypothesis of a very ancient isostatic support for high topography have been put forward by Pedersen et al. (2016) that postulate that the crustal structure that relates to the Caledonian orogeny, 400 Myr old, permitted to maintain the modern Scandinavian margin into which Quaternary fjords were carved at high elevation ever since.

It has been postulated that the northern rim of the Zimbabwe Highland was affected by rift processes in the Late Carboniferous, i.e. at the time of the LPIA (Daly et al., 1989; Nyambe, 1999; Mackintosh et al., 2017; Lopes et al., 2021). The Zimbabwe highland may therefore correspond to the shoulder of this E-W-oriented rift basin upon which the ice masses would have developed and



maintained (Eyles, 1993). Such a high-standing plateau would also explain the presence of a very thin layer of Karoo Supergroup sediments, only a few tens of meters, for 120 Myr of evolution (see also Macintosh et al., 2017).

Together, these lines of evidences strongly suggest that glacial processes reshaped an older surface whose exact nature, most likely polygenic and polyphased, has yet to be unravelled. It must finally be stated that even if little evidence for Ordovician (445 Myr) glacial activity is found over

Southern Africa, scouring action of Ordovician glaciers may have contributed to the shaping of the landsurfaces within a period otherwise dominated by uplift and erosion processes, as indicated by thermochronometrical constraints (Kounov et al., 2013; Baughman and Flowers, 2020, see also Tankard et al., 2009), and therefore poor potential of preservation (Prasicek et al., 2015).

### 5.2.2. Escarpments and valleys: fault-controlled topographic steps localizing glacial
erosion?

The topographic escarpments at the edge of the modern highlands are conventionally interpreted as scarps between planation surfaces, and the valleys incised within these scarps are sometimes taken for pediments and pedivalleys that originated in the Cenozoic break-up system (Guillocheau et al., 2018). In Namibia for example, the escarpment has been suggested to partly relate to Cretaceous rift

processes (Salomon et al., 2015). As it has been shown, these escarpments already existed by LPIA times. In most instances over Southern Africa, the escarpments edging the highlands are surficial expression of crustal-scale faults, either delineating the cratons and their surroundings accreted terranes, intra-cratons faults or Paleozoic rift structures (Figs. 12 & 13; Daly et al., 1989; Tankard et al., 2009a; Begg et al., 2015). These faults may have therefore been reused by glacial processes.

In Namibia, the escarpments edging the Kaoko highland into which paleofjords are carved correspond to faults delineating the Congo Craton to the east and the Kaoko belt to the west (Fig. 12; Goscombe & Gray, 2008). Therefore, considering the crustal structure and the structural evolution of the region prior and during the LPIA, two hypotheses for the genesis of the escarpment and the existence of the high ground are suggested, as summarized in figure 12:

**(1) Hypothesis 1, peneplanation and rejuvenation:** In this hypothesis, relief generated during the Pan-African orogeny that terminated around 480 Myr would have been flattened through peneplanation for 180 Ma, until the LPIA. Immediately before or during the LPIA, tectonic processes such as subsidence of the Kaoko belt that produced the Karoo-aged basins and/or uplift of the Congo craton, reactivated basement structures and faults

inherited from the Panafrican orogen and rejuvenated their surficial expression (Daly et al., 1989; Pysklywec & Mitrovica, 1999; Pysklywec & Quintas, 1999; Tankard et al., 2009). Tectonism and fault reactivation may relate to the extension as Karoo-aged rift systems have been proposed for Northern Namibia (Daly et al., 1989; Clemson et al., 1997, 1999; Aizawa et al., 2000). Such a Late Paleozoic rift system would be signified by



thermochronometric data that indicate a period of enhanced exhumation during the Devonian-Carboniferous after a quiescent period that lasted between the Cambrian and the Devonian, itself following a period of exhumation (Late Neoproterozoic to Cambrian) interpreted as the peneplanation period (curves 1, 2 & 3 in figure 11 of Krob et al., 2020).

**(2) Hypothesis 2, inherited high topography:** The alternative hypothesis is that the
topographic escarpment formed during the Panafrican orogeny, marking the topographic boundary between different tectonic provinces, the Coastal Terrane and the Congo Craton, and persisted since then owing to incomplete peneplanation. This would indicate that the modern relief of the Kaoko is very old, dating to the early Phanerozoic, as it has also been suggested for the Scandinavian Margin by Pedersen et al. (2016) or the Canadian Shield by
(Ambrose, 1964)

These two hypotheses could also have existed jointly, as 'King's (1951) view is that the Great Escarpment is an amalgamation of different palaeotopographic elements' (Aizawa et al., 2000). Moreover, the incision of fjords through the escarpment may have amplified the topography through isostatic uplift (Fig. 12; Medvedev et al., 2018; Pedersen et al., 2019). At this stage, it remains unclear
what controlled the path of local ice flows and its funnelling into what later became the fjords punctured through the escarpment (Kessler et al., 2008). These ice flow paths may correspond to zone of weakness in the underlying basement (e.g., different lithologies, pre-existing weathered areas, faults) or already existing relief possibly fluvial in origin (Visser, 1987a; Jamieson et al., 2008; Benn and Evans, 2010; Livingstone et al., 2017; Bernard et al., 2021); this later hypothesis is particularly
relevant when assuming the Kaoko as an ancient topography which experienced long surface exposure prior to the LPIA (hypothesis 1 above; fig. 12 left).

The southern flank of the Cargonian Highland in South Africa is also marked by an

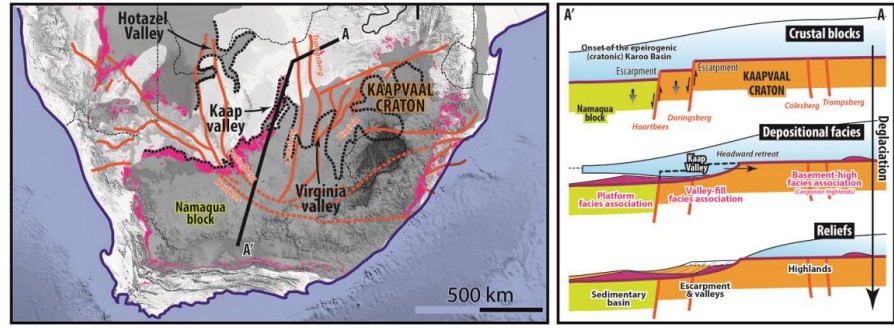

Fig. 13: (a) Structural map of the Main Karoo Basin and the Cargonian Highlands; faults are after Tankard et al. (2009); (b) proposed model for the carving of the Kaap valley (Fig. 6) that corresponds to a headward retreat of the valley due to glacial erosion from the offsetting Doringsberg fault. See text for details.





escarpment into which large glacial valleys are carved (the Kaap and Virginia valleys, fig. 13). These valleys funnelled ice flows and controlled mode of glaciogenic sedimentation. Here, the escarpments

correspond to crustal structures (Fig. 13). Tankard et al. (2009) indicate that subsidence of the MKB started during the LPIA and was initially characterized by vertical motion of rigid crustal blocks that correspond to terranes accreted to the Kaapvaal craton, accommodated by crustal-scale faults between these terranes in an epeirogenic context. On the one hand, in the central MKB, the Virginia glacial valley is fault-controlled as well as the promontory between the Virginia and Kaap valleys (Fig. 13).

The Kaap valley does not seem however to be associated to a fault and may therefore correspond to the headvalley retreat that originated from the escarpment formed by the offset of the Doringsberg fault (Fig. 13). On the other hand, in the eastern Karoo, the Natal escarpment into which smaller glacial valleys are carved (Fig. 10) corresponds to a basement step formed by the Tugela thrust front, delineating the Kaapvaal craton to the north and Natal Province to the South (see figure 13 in Tankard

et al., 2009). We therefore posit that the escarpments and some glacial valleys correspond to surficial expression of basement structures (faults) reactivated immediately before or during the LPIA and exploited and enhanced by glacial erosion.

### 5.2.3. Karoo-aged sedimentary basins: dynamic subsidence or foreland?

Over Southern Africa, thermochronometrical data indicate that the Carboniferous-Permian period, ie

the time of the LPIA, corresponds to a period of tectonic upheaval, namely a transition from compressional regime and uplift and erosion before the LPIA to a regional extensional subsidence and sedimentation and burial after the LPIA (Fig. 11) which likely allowed for preservation of the glacial erosion surfaces (Fig. 1). The pre-LPIA uplift likely resulted from the assembly of Pangea (Veevers et al., 1994; Tankard et al., 2009a). Afterwards, two subsidence mechanisms were proposed for the

existence of the Main Karoo Basin of South Africa. Johnson et al. (1997), Catuneanu (2004), Catuneanu et al. (2005) and Isbell et al. (2008) postulated that subsidence originated from the isostatic and flexural deflection tied to the Cape Orogeny. However, orogeny likely started around the Permian-Triassic boundary, ca. 250 Myr ago, date at which the MKB started to function as a foreland (Linol & DeWit, 2016). Before this tectonic event, it has been proposed that the MKB originated from a

lithospheric deflection pulled down by subduction-driven mantle flow, dynamic subsidence (Pysklywec and Mitrovica, 1999; Pysklywec and Quintas, 1999; Tankard et al., 2009). This dynamic subsidence was first marked by foundering of rigid crustal blocks along pre-existing crustal structures such as faults and then by long-wavelength subsidence (see details in Tankard et al., 2009).




## 6. Conclusions and perspectives

Linley A. Lister (née King, 1936-2016) wrote in 1987 in her treatise on 'The Erosion surfaces of Zimbabwe' that *In many respects the Pre-Karoo landscape of Zimbabwe was remarkably similar to that existing at the present day'*. In the present contribution, we have demonstrated that the same applies to most cratonic areas floored by Archean to Proterozoic rocks over southern Africa. Our findings showcase that late Paleozoic glaciers shaped the Earth surface. Nowadays, after a 300 Ma-long history of burial and exhumation, preserving them from weathering and erosion, these glacial landscapes have been resurrected and characterize the modern landscape of southern Africa.

These observations partly call into question the geomorphological studies carried out in southern Africa, which have interpreted the shelving of the subcontinent as reflecting successive phases of uplift since the break-up of Gondwana (e.g., Partridge and Maud, 1997, 2000; van der Beek et al., 2002; Burke and Gunnell, 2008, Guillocheau et al., 2018). Here, we highlight the importance of volcanic and sedimentary cover in preserving the ancient landforms that must be restored to understand the evolution of the southern African relief.

Whilst our findings apply for southern Africa, similar inferences most likely hold for other cratonic areas that experienced the late Paleozoic Ice Age, as demonstrated by widespread striated surfaces and glacial morphological features or suggested by ancient (Carboniferous-Permian) thermochronometric ages:

- In South America, where paleofjords characterize the edge of Karoo-aged basins (Kneller et al., 2004; Rosa et al., 2016; Tedesco et al., 2016; Mottin et al., 2018; Assine et al., 2018).

- Vast regions of Central Africa host relict glacial morphological features and glaciogenic sediments (Studt, 1913; Dixey, 1937; Boutakoff, 1948; Wopfner and Kreuser, 1986; Ring, 1995; Wopfner and Diekmann, 1996; Catuneanu et al., 2005b) and Carboniferous-Permian thermochronometric ages have been inferred for widespread erosion surfaces such as in Malawi (McMillan et al., 2022, see also Mathian et al., in press)

- Likely in Madagascar, where glacial strata are found at the very base of the Karoo-aged Majunga and Morondava sedimentary basins (Rakotosolofo et al., 1999)

- In India, where Dwyka-equivalent strata lie on the cratonic bedrock (Casshyap and Srivastava, 1987; Dasgupta, 2020)

- In Australia, where vast sedimentary basins bear Dwyka-equivalent strata over cratonic areas (Fielding et al., 2023)

- And in Antarctica where Rolland et al. (2019) postulate the presence of vast glacial landscapes inherited from the LPIA. It may even be conceivable that Cenozoic glacial erosion reused and superimposed glacial reliefs carved during the LPIA (e.g., Carter et al., 2023, this volume).



## Competing interests

The contact author has declared that none of the authors has any competing interests

## Acknowledgements

P. Dietrich and D. Le Heron acknowledge funding from the South Africa–Austria joint project of the National Research Foundation (NRF) of South Africa and the *Österreichischer Austauschdienst* of Austria (OEAD project ZA 08/2019). We dedicate this paper to Alexander du Toit (1878-1948), Henno Martin (1910-1998), Lester King (1907-1989), Linley Lister (1936-2016), Maarten de Wit (1947-2020), and Johann Visser whose work laid the groundwork of the 'ancestral landscapes' of southern Africa.

## Figure captions

Fig. 1: (a) Modern relief of Southern Africa shown by Digital Elevation Model (DEM) from Shuttle Radar Topographic Mission (https://www2.jpl.nasa.gov/srtm/) along with major river networks, international borders and main cities. The transect highlights the high-standing plateaus. (b) Southern Africa with regions of interest discussed in the text shown by red frame. The Archean to Paleoproterozoic Congo, Kaapvaal and Zimbabwe cratons are evidenced by thick orange lines and Karoo-aged sedimentary basins are represented by grey shaded area. The glaciogenic Dwyka group is represented by pink colour. Inset map shows western Gondwana formed by Africa and South America and the four paleohighlands discussed in the text are evidenced in green. Transect displays the thickness and sedimentary succession of Main Karoo Basin (MKB) of South Africa, the glaciogenic Dwyka Group in pink, the glacial erosion surface (wavy pink line) at the base of the Karoo Supergroup and the underlying basement structure (cratons vs. accreted terranes). Transect modified after Johnson et al., 1996 and Karoo-aged basins after Catuneanu et al., 1998.

Fig. 2: (a) DEM of the Kaoko region of Northern Namibia, corresponding to the Kaoko paleohighland. The escarpments, valleys and tongue-shaped troughs discussed in the text are arrowed. Location of the pictures are also indicated. Figure 1b for location. (b) the Gomatum valley corresponds to a fjord carved during the LPIA, later sealed and exhumed in recent times. See Dietrich et al., 2021 for further details. Valley is ca. 2.5 km wide and 550 m deep. (c) A field of *roches moutonnées* and whalebacks characterized by glacial striae and grooves and polished floors covered in places by boulder pavement, evidencing a westward ice movement. Circled geologist for scale, see Le Heron et al. (2024) for details. (d) Striated floor in the Kunene valley, plucking at the joint shows ice movement from east to west. Picture from Martin, 1961; (e) Glacially polished walls and (f) floor in NE Kaoko. Pictures taken by K.E.L. Schalk, geologist Henno Martin for scale, see Miller (2011).



Fig. 3: Geological map indicating the Karoo Supergroup and morphostratigraphic transects across the Kaoko highland, highlighting the morphology of the Kunene, Kaoko, Huab-Ugab regions and the associated glacial valleys and troughs. Etendeka lavas are represented in green, non-glaciogenic Karoo sediments in yellow and Dwyka glaciogenics in pink, or indicated by pink arrows. Black dashed lines on the map represent outlines of exhumed glacial reliefs and valleys; solid purple lines on morphostratigraphic transects represent glacial surfaces and dashed purple lines represent suspected glacial surfaces. Bedrock in grey indicates substrate older than the Karoo Supergroup Note that this colouration is consistently used throughout the manuscript. Fig. 1b for location.

Fig. 4: (a) DEM of the western coastal Kaoko region and (b) morphostratigraphic transect. In the Purros canyon are remnants of glaciogenic sediments, and therefore the canyon is tentatively interpreted here as a relict glacial landform. See Fig. 2 for location.

Fig. 5: (a) DEM of the Windhoek highland (central Namibia), (b) their morphostratigraphic transects. And (c) mosaic picture of the U-shaped Nausgamab valley interpreted by Martin (1961) as a potential glacial valley (see also Miller, 2011). Faupel (1974) reported glaciogenic sediments in the vicinity of this valley; (d) DEM of the Naukluft mountain crosscut by the U-shaped Tsondab valley interpreted by Korn & Martin (1959) and Martin (1961) as a glacial valley. (e) morphostratigraphic transect and (f) picture of the Tsondab valley. Fig. 1b for location.

Fig. 6: DEM of the SW Cargonian Highland (central South Africa and southern Botswana; Fig. 1b for location) and associated geological transects. Widespread Dwyka outcrops in the Kaap valley visible in the landscape interpreted here as an exhumed glacial valley. Diamonds represent kimberlite pipes used for reconstruction in fig. 11. Inset photo: close-up view of the Nooitgedacht glacial pavement (whaleback) in Slater (1932). Circled hammer on the left for scale.

Fig. 7: (a) DEM of the central Cargonian Highland (Johannesburg-Pretoria-Witwatersrand area on the Kaapvaal craton, central South Africa). Fig. 1b for location. The Mooi and Harts river valleys, highlighted by white dashed lines, and the surrounding areas, are interpreted by De Wit (2016) as an exhumed glacial surface. Similarly, the Witbank region to the east, and the Vredefort dome to the south are also interpreted as exhumed glacial surfaces (see text for detail). In between, the Witwatersrand region, the Magaliesburg range, the Pilanesberg dome and the cities of Johannesburg and Pretoria also probably sit on a glacial surface, although further work need to be done to confirm such a hypothesis. (b) View of the Vredefort dome area where the Vaal river valley shows a U-shaped profile reminiscent of glacial erosion. A small portion of the Vaal River floodplain is seen at centre-right (Fig. 4.5 in Gibson & Reimold, 2015)




Fig. 8: (a) DEM of the eastern Cargonian Highland (edge of the Main Karoo Basin; Kaapvaal craton, eastern South Africa) and morphostratigraphic transect highlighting the Tugela valley as an exhumed glacial landscape. Fig. 1b for location. See Details in Dietrich & Hofmann (2019). (b) Bank of the Buffalo River exhuming a glacial valley. Stratified, steeply-dipping rock on the right corresponds to


Archean Pongola Supergroup quartzite into which steep-flanked relief were carved and upon which coarse-grained deposits corresponding to glaciogenics of the Dwyka Group are plastered          .
Although the Dwyka sediments are steeply-dipping on the flank of the (paleo)valley, they become horizontal in the river thalweg. Circled geologist for scale; (c) Landscapes of rolling hills corresponding to an exhumed glacial landscape. The relief is carved into Archean Pongola quartzite


seen in the foreground: topographic lows preserve remnants of glaciogenic sediments whose bulk has been eroded away by recent erosion, resurrecting the glacial landscape. Striated pavements, such as the ones showcase on fig. 7d and 7e, characterize basement floors. (d) Striated floors carved onto volcanic rocks of Archean greenstone belt, plucking of the joint at the foreground indicate an SSW ice movement. (e) Striated and polished glacial floor exposed in a stream, and showcasing a small-scale


*roche moutonnée* behind the circled hammer, evidencing an ice movement to the SSW. The glacial floor is still covered in place by remnants of glaciogenic sediments. (f) A U-shaped trough, 800 m wide and 100 m deep carved by glacial erosion into Archean Pongola quartzites. Remnants of glaciogenic sediments are still present. Picture from Dietrich & Hofmann (2019).


Fig. 9: DEM of the Zimbabwe Highland (central Zimbabwe) and morphostratigraphic transects across the Great Dyke, the Mwanesi Greenstone Belt and the Somabula region. The reader is redirected to Moore & Moore (2006), Moore et al. (2009) and Lister (1986) for further details. Fig. 1b for location.

Fig. 10: (a) Synthesis of glacial paleolandscapes at the scale of Southern Africa. Dark pink indicates attested glacial surface and light pink suspected glacial surfaces whose compilation is based on the


presence of glacial morphological features (see text for details). Dark grey regions are Karoo-aged basins. Exhumed paleo-escarpments are represented by black bold lines and escarpments still buried under sediments are after Visser, 1987a, 1987b. Light orange region corresponds to surficial sediments of the Kalahari Desert, after Haddon (2005). (b) Proposed paleogeographic reconstruction of Southern Africa at the end of the LPIA. Blue-grey areas represent highlands whose names are written in green,


sedimentary basins are represented in dark yellow where attested or light yellow where suspected. Escarpments delineating the highlands from the basins and glacial valleys carved into it are represented as bold solid lines where attested or as dashed lines where suspected (see Visser, 1987a, 1987b). Hills or mountainous regions are also indicated. Region where no data is available mostly





correspond to the Kalahari Desert – see fig. 10a above. Names of glacial valleys and escarpments refer
to those discussed in the text to which the reader is redirected for further details.

Fig. 11: Burial-exhumation history models the Kaoko (Fig. 2), Cargonian (Fig. 6) and Zimbabwe (Fig.
9) highlands. Thermochronological inferences are provided in the graphs, exhumation evidenced from
kimberlites for the Cargonian Highlands are displayed in red and sediment volume accumulated on the
continental margins are showcased in yellow. Raab et al. (2005), Krob et al. (2019) and Margirier et
al. (2019) for the Kaoko; Stanley et al. (2015, 2019, 2021) and Wildman et al., 2015 for central south
Africa and Mackintosh et al. (2017) for central Zimbabwe.

Fig. 12: Structural map of the Kaoko Highland (Fig     s 2 and 3), faults are after Goscombe & Gray
(2008) and two alternative models for the evolution of the escarpments and the associated valleys
before the LPIA, as follows: Hypothesis 1 implies that the relief created at the end of the Panafrican
orogeny around 480 Myr was entirely levelled down (peneplanation) before the LPIA and rejuvenated
owing to vertical crustal movements during or immediately prior the LPIA whose ice flow carved
valleys into it. Hypothesis 2 implies that the relief created by the Panafrican orogeny was only partly
eroded during the time interval between the Panafrican orogeny and the LPIA and was amplified by
glacial processes. The first stage representing the end of the Panafrican orogeny – extension tied to
post-orogenic collapse – is common to both hypothesis and derived from the Goscombe & Gray
(2008). See text for details.

Fig. 13: (a) Structural map of the Main Karoo Basin and the Cargonian Highlands; faults are after
Tankard et al. (2009); (b) proposed model for the carving of the Kaap valley (Fig. 6) that corresponds
to a headward retreat of the valley due to glacial erosion from the offsetting Doringsberg fault. See
text for details.

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
