# Peer review of "The Glacial Paleolandscapes of Southern Africa"

_EGUsphere, 2024_

## Referee Comment (RC1)

[referee-annotated manuscript omitted]

---

## Referee Comment (RC2)

Manuscript review "The Glacial Paleolandscapes of Southern Africa: the Legacy of the Late Paleozoic Ice Age" by Dietrich et al.

Suggestion: Major Revision

The manuscript "The Glacial Paleolandscapes of Southern Africa: the Legacy of the Late Paleozoic Ice Age" by Dietrich et al. explores the influence of Late Paleozoic glaciations on the modern landscapes of Southern Africa, using thermochronological data, geomorphological evidence, and stratigraphy to reconstruct past environments and burial/denudation processes. My field of expertise is in thermochronology, which allows me to assess the interpretations related to the thermal history and exhumation models presented in the paper. However, Sections 1, 2, and 3, which focus on the broader geomorphological context and historical geology of Southern Africa, fall outside my area of specialization, so my comments on these sections are based on general observations.

In general, I find the hypothesis presented in the paper—suggesting that glacial processes during the Late Paleozoic have left a significant and lasting imprint on Southern Africa's landscapes—both interesting and worthy of further exploration. Nonetheless, the paper requires significant revisions, particularly in its reliance on localized data and the absence of broader, quantitative comparisons with modern glacial landscapes, which could better support its hypotheses. Some interpretations, specifically related to the thermochronological data, appear overstated without sufficient supporting evidence or acknowledgment of uncertainties inherent in the methods used. Furthermore, the manuscript would benefit from integrating alternative hypotheses and providing a more balanced discussion of the geomorphological features. Structural improvements, consistent referencing, and the inclusion of supplementary materials are also recommended to enhance clarity and strengthen the paper's arguments.

Major comments are attached to this introduction.

**Major Comments**

1) Sections 1, 2, and 3 fall outside my area of expertise, so I will limit my comments to general impressions, which the editor and authors may assess for relevance. Overall, I believe these sections provide a reasonable summary of previous publications, contextualizing the current landscape as reflective of past glaciogenic processes, though not necessarily confined to the LPIA. However, the reliance on limited data (e.g., photos of outcrops and landscapes) is not particularly convincing in supporting the broader claims, especially for the Zimbabwe Highaland. I suggest incorporating a comparison with modern glacial landscapes for additional context.

   • I think that including supplementary materials that provide more detailed evidence for each locality (e.g., photos with precise locations) would strengthen the argument. In its

current form, the evidence seems quite localized, lacking a broader, more widespread perspective.

- In sections 2 and 3, I also noticed the absence of alternative hypotheses or explanations for these landscapes, which could offer a more comprehensive view. Addressing potential competing interpretations would provide a more balanced discussion of the geomorphological features.

- I also found myself lost at several points in these sections, particularly between lines 368–396 and 533–548. I recommend a revision to make these sections more concise and improve readability.

- Additionally, a more quantitative analysis of the landscape could enhance the argument significantly. This could involve calculating and representing key geomorphological parameters such as slope, relief, ... Comparing these metrics with those from present-day glacial environments would, in my view, lend greater support to the hypotheses outlined by the authors, making their conclusions more convincing.

- I noticed errors in the figure numbers and references. I have highlighted some of these issues (eg., pg. 16. L.399; pg. 18, L.431; Isabell and Cole, 2008; pg. 9 L. 266; ….), but it is the responsibility of the authors to carefully review and correct all misreferences throughout the manuscript.

2) From this point on, I will focus on the thermochronological aspects of the paper, specifically related to the thermal history of the referred crystalline basement (Section 4). In general, there seems to be an over and/or misinterpretation of the thermal paths and the modeling results. The authors need to keep in mind **the limitations of thermochronological modeling** and acknowledge these in the text. I recommend consulting Fox et al. (2020) and Ding (2023), for example, for a more balanced perspective.

**Section 4**

- I disagree with the statement:

> *"Finally, we would like to stress that assessing the controversial exhumation history of this region is beyond the scope of the paper and we objectively provide information we have at hand"* (L. 735).

If you are using these models for your interpretations, you need to explain why these models were chosen over others and justify your methodological decisions.

- It would enhance the paper to include a brief overview of the temperature ranges associated with each thermochronological method (AFT, ZFT, AHe, ZHe). This information would provide readers with a clearer understanding of the applicability and limitations of the various techniques discussed.

**Section 4.1. The Kaoko highland**

Manuscript review "The Glacial Paleolandscapes of Southern Africa: the Legacy of the Late Paleozoic Ice Age" by Dietrich et al.

- This sentence:

  > "Margirier et al. (2019) was used since the early Cretaceous, where Raab et al. (2005) and Krob et al. (2020) were used since the LPIA, although the geological set-up in Krob et al. (2020) may be too restrictive." (L.741)

Is unclear and lacks justification for your choices. I suggest revising it to: "*The thermal modeling of Margirier et al. (2019) for samples from **[insert location]** was selected to represent the thermal history of the region since the early Cretaceous due to **[insert reasons, such as the methodologies employed, quality of the analysis, and presence of reliable constraints]**. Additionally, the thermal modeling paths from Raab et al. (2005) and Krob et al. (2020) were utilized to represent the thermal history of the region since the LPIA (ca. 300 Ma), as they provide **[insert reasons, such as methodological robustness, data quality, and relevant constraints]***.".

- In:

  > "*From the demise of the LPIA until Early Jurassic (190 Ma), i.e. for 110 Ma, thermochronological data indicate a warming of ca. 35°C (Krob et al., 2020), i.e. a burial of 1.4 km considering the thermic gradient described before.*" (L. 747)

it's important to acknowledge the **uncertainty** in the warming estimation. You should note that Krob et al. (2020) used a constraint of surface temperatures between 325-305 Ma, which is hypothetical. This warming of approximately 35°C **was necessary to fit the AFT data** due to this constraint. It would be helpful to briefly explain why these surface temperature constraints are considered reasonable. Without this constraint, the modeled warming would not have been necessary to explain the AFT data.

- The section on the exhumation history of the Kaoko Highlands needs to be reformulated for clarity. Please specify the thermochronological methods used for each assumption regarding the exhumation path. The models from different authors are not contradictory; they are based on various methods and study areas. For example, Margirier et al. (2019) suggest a cooling of approximately 290 °C due to the initial conditions imposed by the model: "Initial conditions for the model are fixed at t = 120 ± 10 Ma and T = 300 ± 50 °C based on Ar-Ar cooling dates (132–130 Ma; Schmitt et al., 2000)." **This estimation applies specifically to this Cretaceous intrusion and should not be generalized to the entire area**.

  **Section 4.2. The Cargonian highland**

- Concerning these parts:

  > "*Thermochronological data are partly contradictory. Wildman et al. (2017) indicate that a linear cooling of 60°C occurred from 350 Myr to today, which would imply 2.4 km of erosion. In line with this, Hanson et al. (2009) and Stanley et al. (2013, 2015) postulate*

Manuscript review "The Glacial Paleolandscapes of Southern Africa: the Legacy of the Late Paleozoic Ice Age" by Dietrich et al.

> on the basis of kimberlite pipes that ca. 1.5-2 km of Karoo sediments have been eroded from the Ghaap plateau, as indicated by the hypabyssal facies of the Makganyene kimberlite cropping out at the surface." L.820

And

> "Contradictory to this model, Baughman and Flowers (2020) and Flowers & Schoene (2010) indicate an abrupt warming of 60°C between 280 and 250 Ma, followed by a quiescent period until 100 Ma." L. 827

I would like to reiterate that these models are not necessarily contradictory. They represent different geological formations and locations, with each model incorporating distinct constraints. It is more appropriate to discuss these models in terms of the methods and data on which each author bases their conclusions, rather than labeling them as conflicting.

Upon reviewing the cited papers, there does not appear to be conclusive evidence for the burial of these paleohighs. The data show that we have Paleozoic apatite fission track (AFT) ages (Wildman et al., 2017), partially eroded Meso-Cenozoic kimberlites, and older zircon (U-Th)/He (ZHe) ages, which Baughman and Flowers (2020) interpret in the context of Precambrian history. The burial events in their models are linked specifically to Paleozoic constraints; if these constraints are accurate, then the proposed burial is necessary.

In conclusion, I recommend focusing the discussion on the Baughman and Flowers (2020) paper and thoroughly justifying the Paleozoic constraint, as it is the primary reason to consider Mesozoic burial. Presenting this analysis will provide a clearer and more cohesive narrative, rather than framing the models as inherently contradictory.

**Section 4.3. The Zimbabwe highland**

- Regarding the section:

> "Thermochronological data from Macintosh et al. (2017) indicate that a ca. 50°C warming occurred, from 300 to ca. 40-25 Ma, corresponding to a burial of 2 km. Compared to the preserved sediment thickness, thermochronological data would imply that a an almost 2 km-thick accumulation of Karoo sediments" L.857

The interpretation of thermochronological data from Macintosh et al. (2017) suggesting a 50°C warming and corresponding burial of 2 km between 300 and 40-25 Ma may be an overstatement. The data from Macintosh et al. (2017) do not conclusively indicate burial. Rather, burial is a possibility, but it remains within the limits of uncertainty given the sensitivity of the data. To summarize, while the data allow for the possibility of slight burial, they do not provide definitive evidence for it. A more cautious interpretation would be that the data are consistent with a possible denudation of up to 2 km since the Paleozoic, and while some degree of burial is feasible, it remains uncertain without additional geological evidence. However, additional evidence is necessary to verify whether basin sediments indeed covered that area.

Manuscript review "The Glacial Paleolandscapes of Southern Africa: the Legacy of the Late Paleozoic Ice Age" by Dietrich et al.

- Regarding the section:

> *"As for the Cargonian Highlands, the offshore stratigraphy of the margin surrounding Southern Africa can provides clues to the history of denudation on land. Thus, the sedimentary isopach map from Baby's (2017) (Figure 7.5 in Baby, 2017 and Figure 7.2 in Ponte, 2018), demonstrates the existence of a Limpopo proto-delta whose watershed may have drained the Zimbabwe region as early as the Lower Cretaceous."* L. 866

The connection between the offshore stratigraphy, the Limpopo proto-delta, and the exhumation of the Cargonian Highlands is unclear. It is essential to explicitly explain how the proto-delta, as indicated by the offshore sedimentary records, is linked to the erosion and denudation processes of the Cargonian Highlands. Currently, the text lacks sufficient detail to establish this relationship. For instance, are the offshore deposits correlated with sediment sourced from the highlands? If so, how does this correlation support the timing and extent of exhumation?

Moreover, Be consistent and precise in citing references. For example, instead of mentioning both "Baby's (2017)" and "Ponte (2018)" separately, ensure that the references are integrated in a cohesive manner, such as: "The sedimentary isopach map presented by Baby (2017) and further discussed by Ponte (2018) suggests…"

3) I find the hypotheses presented in sections 5 and 6 to be both interesting and worthy of discussion. The exploration of Paleozoic landscapes, paleohights, and the behavior of Gondwana's interior in relation to surrounding orogenies are important topics that remain underexplored. These discussions are crucial for understanding intraplate deformation, epeirogeny, sediment flu, craton erosion, and paleoclimate.

While I appreciate the authors' insights, I recommend removing section 5.2.3, as it seems disconnected from the other sections and the overall purpose of the paper. **Furthermore, it is essential to moderate the tone throughout this section.** The authors should clearly state that many questions remain regarding the hypotheses about some current landscapes in southern Africa being shaped by Paleozoic glaciations, to avoid overstating conclusions that are still uncertain.

**Detailed comments (in next round of revision)**

I will provide detailed comments after the authors revise the first draft of the manuscript. These types of comments are time-consuming, and I prefer to address them once the broader issues in the draft have been resolved.

Reviewer:

Manuscript review "The Glacial Paleolandscapes of Southern Africa: the Legacy of the Late Paleozoic Ice Age" by Dietrich et al.

*Ana Carolina Liberal Fonseca*

Ana Carolina Liberal Fonseca

---

## Author Comment (AC2)

**Response to comments**

Reviewer #1: Dr. Fonseca:

The manuscript "The Glacial Paleolandscapes of Southern Africa: the Legacy of the Late Paleozoic Ice Age" by Dietrich et al. explores the influence of Late Paleozoic glaciations on the modern landscapes of Southern Africa, using thermochronological data, geomorphological evidence, and stratigraphy to reconstruct past environments and burial/denudation processes. My field of expertise is in thermochronology, which allows me to assess the interpretations related to the thermal history and exhumation models presented in the paper. However, Sections 1, 2, and 3, which focus on the broader geomorphological context and historical geology of Southern Africa, fall outside my area of specialization, so my comments on these sections are based on general observations.

In general, I find the hypothesis presented in the paper—suggesting that glacial processes during the Late Paleozoic have left a significant and lasting imprint on Southern Africa's landscapes—both interesting and worthy of further exploration. Nonetheless, the paper requires significant revisions, particularly in its reliance on localized data and the absence of broader, quantitative comparisons with modern glacial landscapes, which could better support its hypotheses. Some interpretations, specifically related to the thermochronological data, appear overstated without sufficient supporting evidence or acknowledgment of uncertainties inherent in the methods used. Furthermore, the manuscript would benefit from integrating alternative hypotheses and providing a more balanced discussion of the geomorphological features. Structural improvements, consistent referencing, and the inclusion of supplementary materials are also recommended to enhance clarity and strengthen the paper's arguments.

We warmly thank Dr Fonseca for her thorough review and her insights on thermochronology. The manuscript we present here is based on qualitative exploration of glacial geomorphic and sedimentological features and therefore providing quantitative analysis would be of high value but beyond the scope of the contribution. In particular, we used thermochronological analysis available in the literature to provide of review of the history of burial-exhumation history of the glacial landforms, along with other data (kimberlites, sedimentary thicknesses, budget of erosion and deposition) and it is not our intention, and nor our skills, to discuss uncertainties inherent to thermochronological studies and methods others did. We have however profoundly reworked the section dedicated to 'burial-exhumation history' in order to made it more synthetic, as also asked by reviewer #2 (lines 664 to 695), and stressed the major discrepancies that may exist between the different methods used to reconstruct this history (lines 680-684 and figure 11). On the other hand, as pointed by reviewer, a quantitative comparison with modern glacial landscapes would also be a great value but would require a quantification of glacial erosion for these fossil glacial landscape, which would the scope of a further science article. We have however added some references to modern and quaternary glacial forms about morphometric comparisons (e.g., lines 404-405) Finally, the reviewer suggested to explore alternative hypotheses and a more balanced discussion of the geomorphological features, which was also required by reviewer #2. Accordingly, we have profoundly reshaped the discussion section (lines 765-864) in order to integrate alternate hypotheses about the origin of the fossil landforms observed over southern Africa and their possible pre-glacial origin, and balanced more our statements (and our enthusiasm) about the predominance of glacial landscapes. Some lengthy sections have also been shortened and clarified and referencing has been carefully checked. We believe our manuscript in its new form is clearer and the arguments presented are strengthened. Response to specific comments are attached.

Major Comments

1) Sections 1, 2, and 3 fall outside my area of expertise, so I will limit my comments to general impressions, which the editor and authors may assess for relevance. Overall, I believe these sections provide a reasonable summary of previous publications, contextualizing the current landscape as reflective of past glaciogenic processes, though not necessarily confined to the LPIA. However, the reliance on limited data (e.g., photos of outcrops and landscapes) is not particularly convincing in supporting the broader claims, especially for the Zimbabwe Highland. I suggest incorporating a comparison with modern glacial landscapes for additional context.

☐ I think that including supplementary materials that provide more detailed evidence for each locality (e.g., photos with precise locations) would strengthen the argument. In its current form, the evidence seems quite localized, lacking a broader, more widespread perspective.

We have included all the material we have at hand and we believe geological transect are of high value for justifying the antiquity of the form presented here. For example, the presence of glaciogenic sediment within valleys whose flanks are made of bedrock attest that these valleys date at least back to the LPIA. We have however added different sections where we discuss the possible pre-LPIA origin of some forms we observe and tried to emphasize more the distinction attested and suspected glacial forms.

☐ In sections 2 and 3, I also noticed the absence of alternative hypotheses or explanations for these landscapes, which could offer a more comprehensive view. Addressing potential competing interpretations would provide a more balanced discussion of the geomorphological features.

We have now added a new section in the discussion (5.1.) which explore alternate-complementary explanation for the presence of ancient landforms over southern Africa. In its new form, we hope that the MS and its discussion seems more balanced.

☐ I also found myself lost at several points in these sections, particularly between lines 368–396 and 533–548. I recommend a revision to make these sections more concise and improve readability.

We understand that these sections may look a bit overloaded but they address the key point of describing and interpreting specific glacial forms, along with specific key sites and local names. This MS however intend to provide a comprehensive review of the glacial forms found over southern Africa and therefore we would like to maintain the level of details provided.

☐ Additionally, a more quantitative analysis of the landscape could enhance the argument significantly. This could involve calculating and representing key geomorphological parameters such as slope, relief, ... Comparing these metrics with those from present-day glacial environments would, in my view, lend greater support to the hypotheses outlined by the authors, making their conclusions more convincing.

This would indeed represent a very valuable study to conduct but as pointed out, the MS is already long and therefore such quantitative analysis is outside the scope of the paper. We however keep this relevant suggestion for a further study.

☐ I noticed errors in the figure numbers and references. I have highlighted some of these issues (eg., pg. 16. L.399; pg. 18, L.431; Isabell and Cole, 2008; pg. 9 L. 266; ….), but it is the responsibility of the authors to carefully review and correct all misreferences throughout the manuscript.

We have checked and corrected the references to figures and bibliography.

2) From this point on, I will focus on the thermochronological aspects of the paper, specifically related to the thermal history of the referred crystalline basement (Section 4). In general, there seems to be an over and/or misinterpretation of the thermal paths and the modeling results. The authors need to keep in mind the limitations of thermochronological modeling and acknowledge these in the text. I recommend consulting Fox et al. (2020) and Ding (2023), for example, for a more balanced perspective.

Section 4

This section 4, in the version of the manuscript has been significantly shortened, as suggested by reviewer #2, and the burial-exhumation history is now the focus of section 4.3. (Preservation of the paleolandscape) from line 665 to 695. We have put a focus on the stratigraphic architecture of the Karoo strata and sediment budget along with thermochronology analysis for balance.

☐ I disagree with the statement: "Finally, we would like to stress that assessing the controversial exhumation history of this region is beyond the scope of the paper and we objectively provide information we have at hand" (L. 735). If you are using these models for your interpretations, you need to explain why these models were chosen over others and justify your methodological decisions.

We have removed this sentence as it seems controversial. We would like however to stress the fact that all thermochronological studies that have been conducted in the region we examine have been used for the review we provide in our manuscript. Moreover, as stated above, assessing and discussing in detail the different methods-technics-constraints-initial conditions used in the different references we used is clearly out of the scope of our study and our skills and the reader is redirected to the original papers for further details about the methodologies used. We have however removed statement such as 'contradictory to this model' and 'Thermochronological data are partly contradictory'

☐ It would enhance the paper to include a brief overview of the temperature ranges associated with each thermochronological method (AFT, ZFT, AHe, ZHe). This information would provide readers with a clearer understanding of the applicability and limitations of the various techniques discussed.

Section 4.1. The Kaoko highland

☐ This sentence:

"Margirier et al. (2019) was used since the early Cretaceous, where Raab et al. (2005) and Krob et al. (2020) were used since the LPIA, although the geological set-up in Krob et al. (2020) may be too restrictive." (L.741) Is unclear and lacks justification for your choices. I suggest revising it to: "The thermal modeling of Margirier et al. (2019) for samples from [insert location] was

selected to represent the thermal history of the region since the early Cretaceous due to [insert reasons, such as the methodologies employed, quality of the analysis, and presence of reliable constraints]. Additionally, the thermal modeling paths from Raab et al. (2005) and Krob et al. (2020) were utilized to represent the thermal history of the region since the LPIA (ca. 300 Ma), as they provide [insert reasons, such as methodological robustness, data quality, and relevant constraints].".

☐ In: "From the demise of the LPIA until Early Jurassic (190 Ma), i.e. for 110 Ma, thermochronological data indicate a warming of ca. 35°C (Krob et al., 2020), i.e. a burial of 1.4 km considering the thermic gradient described before." (L. 747) it's important to acknowledge the uncertainty in the warming estimation. You should note that Krob et al. (2020) used a constraint of surface temperatures between 325-305 Ma, which is hypothetical. This warming of approximately 35°C was necessary to fit the AFT data due to this constraint. It would be helpful to briefly explain why these surface temperature constraints are considered reasonable. Without this constraint, the modeled warming would not have been necessary to explain the AFT data.

☐ The section on the exhumation history of the Kaoko Highlands needs to be reformulated for clarity. Please specify the thermochronological methods used for each assumption regarding the exhumation path. The models from different authors are not contradictory; they are based on various methods and study areas. For example, Margirier et al. (2019) suggest a cooling of approximately 290 °C due to the initial conditions imposed by the model: "Initial conditions for the model are fixed at t = 120 ± 10 Ma and T = 300 ± 50 °C based on Ar-Ar cooling dates (132–130 Ma; Schmitt et al., 2000)." This estimation applies specifically to this Cretaceous intrusion and should not be generalized to the entire area.

Section 4.2. The Cargonian highland

☐ Concerning these parts: "Thermochronological data are partly contradictory. Wildman et al. (2017) indicate that a linear cooling of 60°C occurred from 350 Myr to today, which would imply 2.4 km of erosion. In line with this, Hanson et al. (2009) and Stanley et al. (2013, 2015) postulate on the basis of kimberlite pipes that ca. 1.5-2 km of Karoo sediments have been eroded from the Ghaap plateau, as indicated by the hypabyssal facies of the Makganyene kimberlite cropping out at the surface." L.820

And

"Contradictory to this model, Baughman and Flowers (2020) and Flowers & Schoene (2010) indicate an abrupt warming of 60°C between 280 and 250 Ma, followed by a quiescent period until 100 Ma." L. 827 I would like to reiterate that these models are not necessarily contradictory. They represent different geological formations and locations, with each model incorporating distinct constraints. It is more appropriate to discuss these models in terms of the methods and data on which each author bases their conclusions, rather than labeling them as conflicting.

Upon reviewing the cited papers, there does not appear to be conclusive evidence for the burial of these paleohighs. The data show that we have Paleozoic apatite fission track (AFT) ages (Wildman et al., 2017), partially eroded Meso-Cenozoic kimberlites, and older zircon (U-Th)/He (ZHe) ages, which Baughman and Flowers (2020) interpret in the context of

Precambrian history. The burial events in their models are linked specifically to Paleozoic constraints; if these constraints are accurate, then the proposed burial is necessary.

In conclusion, I recommend focusing the discussion on the Baughman and Flowers (2020) paper and thoroughly justifying the Paleozoic constraint, as it is the primary reason to consider Mesozoic burial. Presenting this analysis will provide a clearer and more cohesive narrative, rather than framing the models as inherently contradictory.

Section 4.3. The Zimbabwe highland

☐ Regarding the section: "Thermochronological data from Macintosh et al. (2017) indicate that a ca. 50°C warming occurred, from 300 to ca. 40-25 Ma, corresponding to a burial of 2 km. Compared to the preserved sediment thickness, thermochronological data would imply that a an almost 2 km-thick accumulation of Karoo sediments" L.857 The interpretation of thermochronological data from Macintosh et al. (2017) suggesting a 50°C warming and corresponding burial of 2 km between 300 and 40-25 Ma may be an overstatement. The data from Macintosh et al. (2017) do not conclusively indicate burial. Rather, burial is a possibility, but it remains within the limits of uncertainty given the sensitivity of the data. To summarize, while the data allow for the possibility of slight burial, they do not provide definitive evidence for it. A more cautious interpretation would be that the data are consistent with a possible denudation of up to 2 km since the Paleozoic, and while some degree of burial is feasible, it remains uncertain without additional geological evidence. However, additional evidence is necessary to verify whether basin sediments indeed covered that area.

☐ Regarding the section: "As for the Cargonian Highlands, the offshore stratigraphy of the margin surrounding Southern Africa can provides clues to the history of denudation on land. Thus, the sedimentary isopach map from Baby's (2017) (Figure 7.5 in Baby, 2017 and Figure 7.2 in Ponte, 2018), demonstrates the existence of a Limpopo protodelta whose watershed may have drained the Zimbabwe region as early as the Lower Cretaceous." L. 866 The connection between the offshore stratigraphy, the Limpopo proto-delta, and the exhumation of the Cargonian Highlands is unclear. It is essential to explicitly explain how the proto-delta, as indicated by the offshore sedimentary records, is linked to the erosion and denudation processes of the Cargonian Highlands. Currently, the text lacks sufficient detail to establish this relationship. For instance, are the offshore deposits correlated with sediment sourced from the highlands? If so, how does this correlation support the timing and extent of exhumation? Moreover, be consistent and precise in citing references. For example, instead of mentioning both "Baby's (2017)" and "Ponte (2018)" separately, ensure that the references are integrated in a cohesive manner, such as: "The sedimentary isopach map presented by Baby (2017) and further discussed by Ponte (2018) suggests…"

These sentences and statements have all be removed from the manuscript in its new form as they do not bring much but confusions. Our point in this 'burial-exhumation history' section was to provide a broad, general overview, under the form of a review, of the burial and exhumation the glacial forms have experienced since their carving. We have then therefore chosen to remove details about this and keep a simple statement and figure 11.

3) I find the hypotheses presented in sections 5 and 6 to be both interesting and worthy of discussion. The exploration of Paleozoic landscapes, paleohights, and the behavior of Gondwana's interior in relation to surrounding orogenies are important topics that remain

underexplored. These discussions are crucial for understanding intraplate deformation, epeirogeny, sediment flu, craton erosion, and paleoclimate.

We perfectly agree with this suggestion, also suggested by reviewer #2 and have therefore profoundly redesigned the discussion section to explore further the link between the forms we describe and structural architecture of southern Africa and the tectonic events.

While I appreciate the authors' insights, I recommend removing section 5.2.3, as it seems disconnected from the other sections and the overall purpose of the paper. Furthermore, it is essential to moderate the tone throughout this section. The authors should clearly state that many questions remain regarding the hypotheses about some current landscapes in southern Africa being shaped by Paleozoic glaciations, to avoid overstating conclusions that are still uncertain.

We agree and removed section 5.2.3. and included some statements into the geological setting section. Moreover, we have tried to moderate out one and hyperbole and statements, as also suggested by reviewer #. We hope in its new form the manuscript has better clarity, readability and balance.

Detailed comments (in next round of revision) I will provide detailed comments after the authors revise the first draft of the manuscript. These types of comments are time-consuming, and I prefer to address them once the broader issues in the draft have been resolved.

Ana Carolina Liberal Fonseca
* * *
Reviewer #2: Dr. Hall:

This paper provides a fresh and valuable synoptic perspective on the LPIA in southern Africa. The paper will be of interest to many international geologists and geomorphologists. The focus is on establishing the extent of basement topography inherited in the present relief from the sub-Dwyka tillite unconformity surface. This erosion surface retains in places an impressive array of glacial erosion forms across scales. The extent of these glacial surfaces is claimed to be great – although the paper focusses on valleys rather than extensive erosion surfaces. The persistence of glacial surfaces from the Carboniferous is attributed to the geological recent erosion of Karoo group sedimentary cover.

We warmly thank Dr Adrian Hall for his extensive, detailed and thorough review and relevant suggestions greatly led to the improvement of the manuscript.

The organisation of the paper needs to be significantly improved. The reader is currently led back and forwards in time. I suggest that, for each of the 4 study areas, the new and review information is presented for (i) pre-Dwyka geology, denudation and landforms, (ii) Dwyka glacial landforms and sediments and (iii) timing of re-exposure after post-Karoo erosion and Cretaceous to Cenozoic landscape development. This would allow the Discussion to draw together each strand more much efficiently. The novelty of the paper lies in the results for the pre-Dwyka and Dwyka surfaces.

We have reorganized the paper in order for the information to be presented in chronologic order, so that section 2 (the geological setting, from line 123 to 316) starts with (i) pre-Dwyka events and basement geology, followed by (ii) the record of the ice age and then (iii) post-LPIA evolution and erosion and Cenozoic landscape development. Section dealing with burial-exhumation history has been significantly shortened so that the reader is no longer bring back and forth in time. The discussion has then been improved and reshaped to tackle (i) the possible pre-Dwyka origin of some of the landscape, in order to put a focus on the distinction between pre-Dhyka and glacial processes, as suggested by reviewer (lines 736-832) and (ii) the deciphering between pre-Dwyka forms and post-Dwyka landscape evolution (lines 833-864), once again to stay consistent with the chronological events. We hope in this new form the manuscript has improved clarify and readability.

The paper is currently badly let down by the incorrect use of terminology. Early in the Abstract and the Introduction, planation surfaces are included as landforms of glacial erosion. This is odd because even the voluminous textbook on Glaciers and Glaciation by Benn and Evans makes no mention of planation surfaces as glacial forms. There is good reason for this – as a glance at Wikipedia shows. A planation surface is a large-scale, almost flat surface in geology and geomorphology. Common types of planation surfaces include pediments, pediplains, etchplains, and peneplains. These surfaces are cut across varied rocks and structures. Planation surfaces are mainly formed under fluviatile environments under the control of regional and local base levels. These are not glacial forms. The sub-Dwyka unconformity surface is a *glacial erosion surface*. It has a different form to the younger surfaces described by Guillocheau and many others in southern Africa – these are smooth planation surfaces, mainly etchplains and pediplains - formed by weathering and erosion under fluviatile environments since the Cretaceous. When we look at the cross sections in the paper, we see that in all 4 study areas the sub-Dwyka surface is not flat. It has high relief, with deep (glacial) valleys cutting escarpments, and large hills and basins. The authors should make proper comparisons with Pleistocene glacial erosion surfaces of similar form on cratons in Baffin, Greenland, Scotland, Fennoscandia and Antarctica. All the main topographic elements found on the sub-Dwyka surface are represented in these regions.

There are two main reasons why glacial erosion does not tend to form flat planation surfaces. Firstly, unlike rivers, ice sheets do not respect base level. Glacial overdeepening demonstrates this. Secondly, glacial erosion is selective and strongly influenced by bedrock structures in hard, crystalline bedrock. In Precambrian basement terrain in North America and northern Europe, ice sheet erosion has formed landscapes of areal scouring or cnoc and lochan landscapes. These terrains have high relative relief and roughness, with hills and hills masses standing above deep valleys. The main valleys are excavated along faults and in major fracture zones. These glaciated basement surfaces are not smooth and flat. The effect of glacial erosion is to roughen originally smooth rock surfaces. We see this in the transitions from cold- to warm-based ice zones with decreasing elevations on mountain plateaux and at the edges of the cold-based centres of the former Laurentide and Fennoscandian Pleistocene ice sheets. Old, non-glacial surfaces are progressively stripped of regolith and fracture zones are excavated. Preglacial planation surfaces are dissected and destroyed. Look again at David Sugden's zones of glacial erosion. Mathematical models of Pleistocene glacial erosion on elevated passive margins and in alpine mountains may be in error IMO and in any case have limited applicability to the LPIA in southern Africa.

We perfectly agree and have removed mention to 'glacial planation surfaces' throughout the manuscript and replaced them by adequate terms for each study sites, such as selective linear

erosion or areal scouring. Moreover, we have paid more attention to the proper descriptions of geomorphic features to make them match what one could see in Canada or Scotland, such as field of roches moutonnées, depositional crag-and-tails, or over modern continental platform such as cross-shelf troughs, and supported by consistent bibliography.

In the text it is made clear that erosion during the LPIA is thought to represent an important phase of denudation, but the paper does not assess the depth of glacial erosion in basement. The depth of Pleistocene glacial erosion of shields has a long history of debate – similar questions have been asked about erosion of cratons during Snowball Earth and can be asked about erosion depths in the LPIA in southern Africa. Given the long duration of LPIA glaciation, pre-LPIA landform inheritance is telling us something about the effectiveness of ice sheet erosion on cratons.

We are aware of the long debate about the depth of Pleistocene glacial erosion and similar questions would undoubtedly arise in the case of the LPIA erosion. However, although the relevance of this question, we think quantifying the amount of LPIA glacial erosion is beyond the scope of the present manuscript. As the new section 5.1 (lines 736-832) is specifically dedicated to the preservation of pre-LPIA erosion forms, we have mentioned that this very preservation of pre-glacial form would suggest limited glacial erosion, at least locally, which could be used to assess and quantify glacial erosion (lines 749-751).

Further issues in terminology include the following. The simplest term to describe the buried sub-Dwyka surface is as an *unconformity*. Clear evidence that the *sub-Dwyka unconformity surface* is locally preserved beyond sedimentary cover is provided by glacial landforms. But a clearer distinction needs to be made with the *sub-Karoo unconformity surface* where the Dwyka is missing and the cover is younger (Triassic). In both cases, an unconformity has been *re-exposed* and so revealed *inherited relief*. *Re-exposure* is included in *exhumation,* but the latter is a wider term that describes the relative movement of a parcel of rock towards the Earth's surface during denudation or through melt or tectonics. Other terms used in the ms like *resurrected* and *rejuvenated* are not appropriate. I suggest that re-exposure is used when referring to unconformities in this paper and that exhumation is used in other contexts, including thermochronology. On recent re-exposure, the modern rock surface and the sub-Dwyka unconformity surface remain at the same erosion level. This is *inherited relief*. Where glacial rocks and minor glacial landforms are not preserved but the wider topography retains an overall glacial form then we have *persistent relief.* Here, the modern rock surface stands metres to perhaps tens of metres below the former sub-Dwyka or sub-Karoo unconformities; the present erosion level is similar to but below the former unconformity surface. Where no inherited or persistent Dwyka or Karoo relief is preserved then the modern rock surface is at a greater depth below the former sub-Dwyka surface. That depth difference is hard to estimate and likely varies. The present surface forms may have been reshaped entirely by geologically recent processes. This latter point is important for understanding the development of the present surfaces on uplands between LPIA glacial troughs than were exposed to renewed weathering and erosion in the Neogene or longer.

This terminology inconsistency has been carefully checked throughout and *re-exposure* has been used when referring to unconformities and *exhumation* has been used when referring to thermochronology and other, wider uses (e.g., line 713). The other terms such as resurrected and rejuvenated have been suppressed.

The terms and related concepts are important for a correct understanding of where the sub-Dwyka and sub-Karoo unconformity surfaces and landforms fit into the long term history of denudation and relief development on the cratons of southern Africa. Significantly, the authors make little mention of pre-Dwyka unconformities even though these are described in the geological literature. For example, in Namibia, the sub-Ediacaran unconformity is an extensive low inclination planar basement unconformity that locally lies close to the LPIA and the present erosion level. Palaeogeographic reconstructions show large parts of southern Africa exposed as land in the Cambrian. Also, De Wit describes LPIA eskers resting on a *polyphase erosional surface*. Lister describes how the sub-Karoo surface in Zimbabwe is indeed extensive, but it is diachronous, has glacial and non-glacial surface forms. and was widely modified on uplands by Cenozoic erosion. Such comparisons are important because earlier writers like King and Lister clearly thought that non-glacial processes had contributed significantly to cratonic denudation before the LPIA. The pre-Dwyka denudation history in southern Africa is little known and warrants more attention. The authors recognise some major pre-LPIA landforms but do not carefully assess their character, form and origin in relation to geological structures or the extent to which these have been reshaped by glacial erosion. Too little mapped geological information is presented.

We have now designed and entire new section (5.1. Pre-LPIA evolution: existing reliefs amplified by glacial erosion? lines 736-834) to address this crucial question of the pre-glacial landform heritage later reused by glacial erosion processes. We segmented this section into two sections dedicated to (i) the surficial expression of basement structures, and (ii) the pre-glacial alluvial reliefs, both underlined by geological information (crustal structure, sediment provenance, shape of the reliefs etc.).

The main results concern the extent of the LPIA (Dwyka) glacial surfaces in the present relief. Results should be more clearly separated from review material. More information should be given on these glacial landforms, especially the supposed "planation surfaces". The only morphological criterion for recognition of glacial valleys appears to be U-shaped cross profiles. Why not compare and contrast N Namibia with E Greenland or similar? Much of the reviewed and new information presented here is about glacial valleys. This is understandable – sedimentary rocks in topographic depressions are often the last to be eroded out. The sub-Devonian valleys around the Cairngorms, Scotland, provide an example. However, the Devonian hills and mountains that once separated the Cairngorm valleys no longer exist, due to later erosion. Similarly, LPIA glacial valleys show that ice sheets may have covered the uplands that separate them but whether the present surfaces of the uplands carry inherited, persistent or younger forms often remains uncertain. This depends on the presence of Dwyka outliers or clear glacial bedforms and these appear to be of more restricted distribution on uplands between valleys than the authors imply. A key point seems to be that systems of LPIA glacial valleys are more widespread than previously thought. In turn, this indicates that pre-Dwyka uplands and escarpments remain as important elements of the present relief, as Lister recognised nearly 40 years ago.

The post-Dwyka history of denudation is reviewed at length in the context of the exhumation of the unconformity, but this is all review material and not new. This can be shortened, but some attention should be given to *when* sub-Karoo unconformities were re-exposed in different locations.

We have significantly shortened the review section about the burial-exhumation (from 170 lines to 30 lines, now section 43.3, from line 664 to 694). Moreover, assessing the exact timing of

re-exposure of each study site is highly challenging, as thermochronology and other data used (all reviews, as pointed out by the reviewer) to assess the burial-exhumation history are sometimes contradictory, and often not precise enough in both time and space. We have however tried to provide an overview of the timing of re-exposure for three emblematic sites (Kaoko, Cargonian and Zimbabwe Highlands) on figure 11.

It is not acceptable to move from the main result *"over Southern Africa, an area of ca. 71.000 km² consists of exhumed glacial landscapes and 360.000 km² correspond to suspected glacial landscapes, which together correspond to ca. 10% of the total area of the region"* to the interpretation of "*vast and well-preserved glacial paleoreliefs*" and "*this modern morphology of Southern Africa is in fact largely inherited from glacial erosion associated to the Late Paleozoic Ice Age (LPIA).*" Focus on what is attested, rather than suspected, and acknowledge uncertainty throughout. Just because a surface carries glacial forms does not mean that there has been deep glacial erosion. Consider southern Sweden where lowering of the sub-Cambrian uniformity by Pleistocene glacial erosion is of the order of 20 m.

We perfectly agree and stressed the fact lines 749-751 that the persistence pre-glacial forms suggest little glacial erosion. We have tried to avoid the use of superlative (e.g., from line 598 to 622) such as the one highlighted above and been more cautious in the distinction of attested and suspected relief, as suggested by the reviewer (from lines 318-592).

My recommendation is for major revision. Fix the organization of the text and its terminology. Add information to the maps (geology, structures, ice flow, ice margins). Such revision need not be very time consuming but requires a more critical approach to what the evidence actually shows. Provide more reference to comparable Pleistocene landscapes of glacial erosion on shields. In the present ms, there a lot of loose expression, dubious logic, and occasional hyperbole in a paper that should have been scrutinised more carefully by its 13 co-authors. I have highlighted the worst cases in my detailed comments.

Once again, we thank Dr Hall for his detailed and thorough review and followed most of his comments and suggestions throughout the ms in order to remove superlative, inappropriate expressions and loose logic. Finally, we have a particular attention to refer to correct glacial erosion processes and resulting landforms.

---

## Referee Report (RR1)

Manuscript review "The Glacial Paleolandscapes of Southern Africa: the Legacy of the Late Paleozoic Ice Age" by Dietrich et al.

Suggestion: Minor Revision

The revised manuscript, "The Glacial Paleolandscapes of Southern Africa: The Legacy of the Late Paleozoic Ice Age" by Dietrich et al., shows significant improvement. The hypothesis is now presented with greater caution and is well-supported by arguments, while the core message of the paper remains intact. The thermochronological data and modelling substantiate their proposals on landscape evolution (as well as other hypotheses) without invalidating their main conclusions. Most of the problematic statements were removed. I have included minor revisions in the attached text. I sincerely thank the authors and editors for the opportunity to review this work.

**Minor comments**
- Figs. 3, 4, 5, 6, 7, 8, 9 = The maps need grid.

Abstract

- Ln. 36 = vast surfaces are exhumed glacial landscapes tied to the LPIA.
  Change to = vast surfaces might be exhumed glacial landscapes tied to the LPIA.

- Ln. 45-48 = Glacial landforms have survived over hundreds of million years. This preservation and modern exposure were achieved through burial under piles of Karoo sediments and lavas over ca. 120 to 170 million years and a subsequent exhumation since the middle Mesozoic owing to the uplift of Southern Africa.
  Change to = To explain how the glacial landscape has survived for such an extended period, we argue that its preservation and modern exposure may be attributed to burial under substantial layers of Karoo sediments and lavas for approximately 120 to 170 million years, followed by its exhumation since the middle Mesozoic, linked to the uplift of Southern Africa.

- Ln. 57-59 = Also, some hill or mountain ranges already existed by LPIA times, likely an expression of Pan-African orogenic belts, whose relief was either reactivated or persisted since then, and was ultimately modelled by glacial erosion. We finally propose that a network of alluvial valleys existed before the LPIA, as southern Africa experienced a long period of exhumation and erosion, and that later served as funneling ice flows from highlands to lowlands.
  Change to = Additionally, some hill or mountain ranges may have already existed during LPIA times, potentially reflecting remnants of Pan-African orogenic belts. Whether these features were later reactivated or persisted unchanged since that time is uncertain, but they were shaped by glacial erosion. We further propose that a network of pre-existing alluvial valleys could have existed before the LPIA, possibly formed during an extended period of exhumation and erosion in southern Africa. These valleys may have later

facilitated ice flow from highlands to lowlands, although the extent and configuration of such features remain speculative.

- Ln. 63 = These exhumed pre-LPIA landforms may in some cases be taken for
  Change for = The exhumed….

1. Introduction

   No suggestions

2. The relief and geology of southern Africa and the record of ice ages

   - Ln. 147 = Flowers, 2019) until ca. 300 Ma, as expressed by thermochronology cooling. Localised subsidence
     Change to = Flowers, 2019) until ca. 300 Ma, as expressed by thermochronology data. Localised subsidence
   - Ln. 213 = Catuneanu et al., 2005; Griffis et al., 2018, 2019a, 2021). The Dwyka Groupis extensively present in
     Change to = Catuneanu et al., 2005; Griffis et al., 2018, 2019a, 2021). The Dwyka Group is extensively present in
   - Ln. 280 = with other inferences (assessment of sediment routing, characterization of kimberlite pipes etc.) allowed
     Change to = with other inferences, e.g., assessment of sediment routing, characterization of kimberlite pipes, allowed …

3. Glacial paleorelief of Southern Africa

   - Correct the numbering! 3., 3.1, 3.3, and 3.4.
   - Ln. 416 = neoproterozoic
     Change to Neoproterozoic
   - Ln. 437 = Dwyka sediments occur within the valley thalweg (Fig. 5d, e & f). Geological map also indicates
     Change to (double check for similar use of &, e.g., Ln 446) = Dwyka sediments occur within the valley thalweg (Fig. 5d, e, and f). Geological map also indicates

4. SYNTHESIS AND IMPLICATION: THE GLACIAL PALEOLANDSCAPES OF SOUTHERN AFRICA AND THEIR PRESERVATION
   - Ln. 657 = functioning of the escarpment passive margin (see above section 2.4; Braun et al., 2014; Braun, 2018a),
     Change to = functioning of the escarpment passive margin (see section 2.4; Braun et al., 2014; Braun, 2018a),
   - Ln. 684-686 = For the need of the reconstruction of the burial-exhumation history from thermochronometrical data (apatite and zircon fission tracks, (U-684 Th-Sm)/He on

apatite), geothermal gradients of 25°C.km-1 are assumed for the Kaoko, and Zimbabwe and Cargonian highlands (Mackintosh et al., 2019; Macgregor et al., 2020).

Change to = To reconstruct the burial-exhumation history using thermochronometric data—such as apatite and zircon fission tracks, as well as (U-Th-Sm)/He analyses on apatite—geothermal gradients of 25°C/km are assumed for the Kaoko, Zimbabwe, and Cargonian Highlands (Mackintosh et al., 2019; Macgregor et al., 2020).

- Ln. 692-656 = For example, over the Kaoko highland (Fig. 11a), thermochronometrical data of Margirier et al. (2019) indicate that a significant cooling of ca. 200°C occurred between 130 and 100 Ma while thermochronometrical data of Raab et al. (2005) rather indicate a cooling of 120°C between 100 and 65 Ma.

The statements are not necessarily controversial. Margirier et al. (2019) describe cooling of approximately 200°C between 130 and 100 Ma, indicating a significant thermal event during this time. Raab et al. (2005), on the other hand, report cooling of about 120°C between 100 and 65 Ma, suggesting another phase of cooling. These observations could represent sequential cooling events, each associated with distinct geological processes.

However, I understand the point you are making. **These cooling events are not solely about the total amount of cooling** but rather **about when the rocks passed through a specific temperature range**. It would be better to rewrite the sentence as follows:

"For example, over the Kaoko highland (Fig. 11a), thermochronometric data from Margirier et al. (2019) and Raab et al. (2005) indicate different times when these rocks passed through the temperature range of 120–60°C. The former suggests significant cooling of approximately 200°C between 130 and 100 Ma, while the latter indicates cooling of about 120°C between 100 and 65 Ma."

5. Discussion

Ln.859 = If you are using the abbreviation LPIA do it systematically. I notice that it also happened with other abbreviations.

6. Conclusion

This is the part of the manuscript I find the least satisfactory. I believe the conclusions could be significantly more concise, perhaps presented as bullet points or in a more structured format.

The evidence for the preservation of glacial landscapes in other study areas was not discussed in detail within the manuscript itself, so I question why it is introduced in the conclusions. While I understand this section aims to provide perspectives, the connection is not entirely clear to me. Additionally, at least in Brazil, I have not encountered references to the preservation of these landscapes. To my knowledge, there are only a few examples of striated pavements, and perhaps I am unaware of more significant evidence (though this could reflect my own limited knowledge).

Manuscript review "The Glacial Paleolandscapes of Southern Africa: the Legacy of the Late Paleozoic Ice Age" by Dietrich et al.

I suggest restructuring this section to succinctly summarize the main evidence supporting your hypothesis and explain how the preservation of these landscapes occurred. Toward the end, you could add a statement along the lines of: "Other regions may also exhibit similar preservation of glacial landscapes, such as in [specific examples or locations]."

Reviewer:

Ana Carolina Liberal Fonseca

---

## Author Response (AR2)

Géosciences Rennes
UMR 3118
CNRS

**Pierre D**IETRICH
PhD

Rennes, February 14th, 2025

**Object:** Submission of a reviewed manuscript to *Earth Surface Dynamics*

Dear editors,

We are pleased to resubmit the manuscript #2024-467 entitled: **The glacial paleolandscapes of Southern Africa: The legacy of the Late Paleozoic Ice Age** by P. Dietrich et al. for consideration in the Special Issue *'Icy landscapes of the Past'* of *Earth Surface Dynamics*.

We have carefully considered the minor comments and suggestions made by the reviewers and have revised our manuscript accordingly. We have addressed most of Dr Ana Fonseca's comments and suggestions, as detailed in the attached files containing the revised manuscript and detailed responses to the reviewers' comments.

The authors have not had any discussion regarding this manuscript with editors of *Earth Surface Dynamics* and are unaware of any conflicts of interest. None of our findings have been published or submitted for review in another journal.

We look forward to your editorial decision.

Yours sincerely,
Pierre Dietrich (corresponding authors)

263 Av. du G\al Leclerc
Bâtiment 15, campus de Beaulieu
Université de Rennes 1
35042 Rennes Cedex
France

F (33) 06 52 95 45 39
pierre.dietrich@univ-rennes1.fr

[Figure]

**Response to comments by Ana C. Fonseca**

Manuscript review "The Glacial Paleolandscapes of Southern Africa: the Legacy of the Late Paleozoic Ice Age" by Dietrich et al.

Suggestion: **Minor Revision**

The revised manuscript, "The Glacial Paleolandscapes of Southern Africa: The Legacy of the Late Paleozoic Ice Age" by Dietrich et al., shows significant improvement. The hypothesis is now presented with greater caution and is well-supported by arguments, while the core message of the paper remains intact. The thermochronological data and modelling substantiate their proposals on landscape evolution (as well as other hypotheses) without invalidating their main conclusions. Most of the problematic statements were removed. I have included minor revisions in the attached text. I sincerely thank the authors and editors for the opportunity to review this work.

**We warmly thank Dr. Fonseca for her second review and encouraging remarks.**

**Minor comments**

• Figs. 3, 4, 5, 6, 7, 8, 9 = The maps need grid.

**Done**

**Abstract**

• Ln. 36 = vast surfaces are exhumed glacial landscapes tied to the LPIA.

Change to = vast surfaces might be exhumed glacial landscapes tied to the LPIA.

**As we demonstrate that many modern forms are glacial fossil landforms, we believe using the initial formulation best fit our point here.**

• Ln. 45-48 = Glacial landforms have survived over hundreds of million years. This preservation and modern exposure were achieved through burial under piles of Karoo sediments and lavas over ca. 120 to 170 million years and a subsequent exhumation since the middle Mesozoic owing to the uplift of Southern Africa.

Change to = To explain how the glacial landscape has survived for such an extended period, we argue that its preservation and modern exposure may be attributed to burial under substantial layers of Karoo sediments and lavas for approximately 120 to 170 million years, followed by its exhumation since the middle Mesozoic, linked to the uplift of Southern Africa.

**Agreed and reworded accordingly**

• Ln. 57-59 = Also, some hill or mountain ranges already existed by LPIA times, likely an expression of Pan-African orogenic belts, whose relief was either reactivated or persisted since then, and was ultimately modelled by glacial erosion. We finally propose that a network of alluvial valleys existed before the LPIA, as southern Africa experienced a long period of exhumation and erosion, and that later served as funneling ice flows from highlands to lowlands.

Change to = Additionally, some hill or mountain ranges may have already existed during LPIA times, potentially reflecting remnants of Pan-African orogenic belts. Whether these features were later reactivated or persisted unchanged since that time is uncertain, but they were shaped by glacial erosion. We further propose that a network of pre-existing alluvial valleys could have existed before the LPIA, possibly formed during an extended period of exhumation and erosion in southern Africa. These valleys may have later facilitated ice flow from highlands to lowlands, although the extent and configuration of such features remain speculative.

**Reworded**

• Ln. 63 = These exhumed pre-LPIA landforms may in some cases be taken for
Change for = The exhumed….
**Done**

**1. Introduction**
No suggestions

**2. The relief and geology of southern Africa and the record of ice ages**
• Ln. 147 = Flowers, 2019) until ca. 300 Ma, as expressed by thermochronology cooling. Localised subsidence
Change to = Flowers, 2019) until ca. 300 Ma, as expressed by thermochronology data. Localised subsidence
**Done**

• Ln. 213 = Catuneanu et al., 2005; Griffis et al., 2018, 2019a, 2021). The Dwyka Groupis extensively present in
Change to = Catuneanu et al., 2005; Griffis et al., 2018, 2019a, 2021). The Dwyka Group is extensively present in
**Done**

• Ln. 280 = with other inferences (assessment of sediment routing, characterization of kimberlite pipes etc.) allowed
Change to = with other inferences, e.g., assessment of sediment routing, characterization of kimberlite pipes, allowed …
**Done**

**3. Glacial paleorelief of Southern Africa**
• Correct the numbering! 3., 3.1, 3.3, and 3.4.
**Done**

• Ln. 416 = neoproterozoic
Change to Neoproterozoic
**Done**

• Ln. 437 = Dwyka sediments occur within the valley thalweg (Fig. 5d, e & f). Geological map also indicates
Change to (double check for similar use of &, e.g., Ln 446) = Dwyka sediments occur within the valley thalweg (Fig. 5d, e, and f). Geological map also indicates
**Done and checked throughout**

**4. SYNTHESIS AND IMPLICATION: THE GLACIAL PALEOLANDSCAPES OF SOUTHERN AFRICA AND THEIR PRESERVATION**
• Ln. 657 = functioning of the escarpment passive margin (see above section 2.4; Braun et al., 2014; Braun, 2018a),
Change to = functioning of the escarpment passive margin (see section 2.4; Braun et al., 2014; Braun, 2018a),
**Done**

• Ln. 684-686 = For the need of the reconstruction of the burial-exhumation history from thermochronometrical data (apatite and zircon fission tracks, (U-Th-Sm)/He on apatite), geothermal

gradients of 25°C.km-1 are assumed for the Kaoko, and Zimbabwe and Cargonian highlands (Mackintosh et al., 2019; Macgregor et al., 2020).
Change to = To reconstruct the burial-exhumation history using thermochronometric data—such as apatite and zircon fission tracks, as well as (U-Th-Sm)/He analyses on apatite—geothermal gradients of 25°C/km are assumed for the Kaoko, Zimbabwe, and Cargonian Highlands (Mackintosh et al., 2019; Macgregor et al., 2020).

**Reworded**

• Ln. 692-656 = For example, over the Kaoko highland (Fig. 11a), thermochronometrical data of Margirier et al. (2019) indicate that a significant cooling of ca. 200°C occurred between 130 and 100 Ma while thermochronometrical data of Raab et al. (2005) rather indicate a cooling of 120°C between 100 and 65 Ma.

The statements are not necessarily controversial. Margirier et al. (2019) describe cooling of approximately 200°C between 130 and 100 Ma, indicating a significant thermal event during this time. Raab et al. (2005), on the other hand, report cooling of about 120°C between 100 and 65 Ma, suggesting another phase of cooling. These observations could represent sequential cooling events, each associated with distinct geological processes.

However, I understand the point you are making. **These cooling events are not solely about the total amount of cooling** but rather **about when the rocks passed through a specific temperature range**. It would be better to rewrite the sentence as follows:

"For example, over the Kaoko highland (Fig. 11a), thermochronometric data from Margirier et al. (2019) and Raab et al. (2005) indicate different times when these rocks passed through the temperature range of 120–60°C. The former suggests significant cooling of approximately 200°C between 130 and 100 Ma, while the latter indicates cooling of about 120°C between 100 and 65 Ma."

**Agreed and reworded accordingly**

**5. Discussion**
Ln.859 = If you are using the abbreviation LPIA do it systematically. I notice that it also happened with other abbreviations.

**Done and checked throughout**

**6. Conclusion**
This is the part of the manuscript I find the least satisfactory. I believe the conclusions could be significantly more concise, perhaps presented as bullet points or in a more structured format.
The evidence for the preservation of glacial landscapes in other study areas was not discussed in detail within the manuscript itself, so I question why it is introduced in the conclusions. While I understand this section aims to provide perspectives, the connection is not entirely clear to me. Additionally, at least in Brazil, I have not encountered references to the preservation of these landscapes. To my knowledge, there are only a few examples of striated pavements, and perhaps I am unaware of more significant evidence (though this could reflect my own limited knowledge). I suggest restructuring this section to succinctly summarize the main evidence supporting your hypothesis and explain how the preservation of these landscapes occurred. Toward the end, you could add a statement along the lines of: "Other regions may also exhibit similar preservation of glacial landscapes, such as in [specific examples or locations]."

**Our intention here is to summarize our main findings to provide a comprehensive framework of the fossil forms of Southern Africa. Moreover, large-scale fossil glacial landscapes such as paleofjords are abundant and well-documented in South America, particularly in Brazil (e.g., Tedesco et al., 2016), but also elsewhere over Gondwana. Therefore, reviewing these fossils forms is beyond the scope of this manuscript and this ultimate section that serve as a perspective intends to advertise the reader that similar fossil landscapes exist and therefore similar findings may apply.**

---

## Author Response (AR3)

Rennes, March 24th, 2025

Géosciences Rennes
UMR 3118
CNRS

**Pierre DIETRICH**
PhD

**Object:** Submission of a reviewed and corrected manuscript to *Earth Surface Dynamics*

Dear editors,

We are pleased to resubmit the manuscript #2024-467 entitled: **The glacial paleolandscapes of Southern Africa: The legacy of the Late Paleozoic Ice Age** by P. Dietrich et al. for consideration in the Special Issue *'Icy landscapes of the Past'* of *Earth Surface Dynamics*.

We have made the technical corrections requested by the Associate Editor and added a section 'author contribution' by using initials for authors' name.

The authors have not had any discussion regarding this manuscript with editors of *Earth Surface Dynamics* and are unaware of any conflicts of interest. None of our findings have been published or submitted for review in another journal.

We look forward to your editorial decision.

Yours sincerely,
Pierre Dietrich (corresponding authors)

263 Av. du Gal Leclerc
Bâtiment 15, campus de Beaulieu
Université de Rennes 1
35042 Rennes Cedex
France

F (33) 06 52 95 45 39
pierre.dietrich@univ-rennes1.fr